# Observation of a non-reciprocal skyrmion Hall effect of hybrid chiral skyrmion tubes in synthetic antiferromagnetic multilayers

Takaaki Dohi [1,2,10] ✉, Mona Bhukta [2,10], Fabian Kammerbauer [2,10], Venkata Krishna Bharadwaj[2], Ricardo Zarzuela [2] ✉, Aakanksha Sud [1,3], Maria-Andromachi Syskaki [2,4], Duc Minh Tran [2], Thibaud Denneulin [5], Sebastian Wintz[6,7], Markus Weigand [6,7], Simone Finizio [8], Jörg Raabe [8], Robert Frömter [2], Rafal E. Dunin-Borkowski [5], Jairo Sinova [2] & Mathias Kläui [2,9] ✉

A hybrid chiral skyrmion tube is a well-known example of a 3D topological spin texture, exhibiting an intriguing chirality transition along the thickness direction. This transition progresses from left-handed to right-handed Néel-type chirality, passing through a Bloch-type intermediate state. Such an exotic spin configuration potentially exhibits distinctly different dynamics from that of the common skyrmion tube that exhibits a homogeneous chirality; yet these dynamics have not been ascertained so far. Here, we reveal the distinct features of current-induced dynamics that result from the hybrid chiral skyrmion tube structure in synthetic antiferromagnetic (SyAFM) multilayers. Strikingly, the SyAFM hybrid chiral skyrmion tubes exhibit a non-reciprocal skyrmion Hall effect in the flow regime. The non-reciprocity can even be tuned by the degree of magnetic compensation in the SyAFM systems. Our theoretical modeling qualitatively corroborates that the non-reciprocity stems from the dynamic oscillation of skyrmion helicity during its current-induced motion. The findings highlight the critical role of the internal degrees of freedom of these complex skyrmion tubes for their current-induced dynamics.

Non-reciprocity, namely the directional dynamics in physical systems, is not only of major importance for fundamental physical questions but also ubiquitously used in our technical world e.g., the ratchet mechanism or rectification. In condensed-matter physics, the non-reciprocal transport of quasi-particles is of particular interest[1], and the exploration of non-reciprocal phenomena is key for realizing new functionality and technological breakthroughs in modern electronics[2–6].

Topological spin textures that emerge in magnetic materials and behave as quasi-particles[7–18] have been extensively investigated in the realm of electronics[19–22] for next-generation devices. Recent studies have unveiled the potential of two-dimensional (2D) topological spin

[1]Laboratory for Nanoelectronics and Spintronics, Research Institute of Electrical Communication, Tohoku University, Sendai, Japan. [2]Institut für Physik, Johannes Gutenberg-Universität Mainz, Mainz, Germany. [3]Frontier Research Institute for Interdisciplinary Sciences, Tohoku University, Sendai, Japan. [4]Singulus Technologies AG, Kahl am Main, Germany. [5]Ernst Ruska-Centre for Microscopy and Spectroscopy with Electrons, Forschungszentrum Jülich, Jülich, Germany. [6]Max Planck Institute for Intelligent Systems, Stuttgart, Germany. [7]Helmholtz-Zentrum Berlin für Materialien und Energie GmbH, Berlin, Germany. [8]Swiss Light Source, Paul Scherrer Institut, Villigen, PSI, Switzerland. [9]Center for Quantum Spintronics, Norwegian University of Science and Technology, Trondheim, Norway. [10]These authors contributed equally: Takaaki Dohi, Mona Bhukta, Fabian Kammerbauer. ✉e-mail: takaaki.dohi.e5@tohoku.ac.jp; rzarzuela@uni-mainz.de; klaeui@uni-mainz.de

textures, particularly skyrmions in thin-film systems, for applications in probabilistic computing[23–26] and other devices[27]. This promise is attributed to the inherently strong fluctuations of spins in thin-film systems, leading to enhanced thermally-activated diffusive motion[28]. However, such 2D spin structures exhibit fully reciprocal dynamics as there are no symmetry-breaking mechanisms available due to their limited complexity. On the other hand, three-dimensional (3D) topological spin textures beyond 2D magnetic skyrmions, such as skyrmion tubes[11,12,29,30], cocoons[31], or hopfions[32,33] offer promising perspectives for devices requiring robust and stable operation[34,35]. It has been shown that the more complex dynamics of such 3D spin configurations enable novel functionality such as triggering extremely non-linear electromagnetic responses, which leads to advanced skyrmionic applications, including skyrmion-based logic with diodes[36,37] and neuromorphic computing[38,39], where one of the indispensable crucial components is the non-reciprocal dynamics of the topological objects.

In thin-film multilayer systems with an interfacial Dzyaloshinskii-Moriya interaction (DMI), a hybrid chiral skyrmion tube[40,41] is an archetypal example of a 3D topological spin texture. Hybrid chiral skyrmion tubes can emerge due to the energy competition between long-range dipole interaction and interfacial DMI. Their structure uniquely exhibits twisting of the helicity across the thickness direction, which can result in intrinsically distinct dynamics compared to that of conventional skyrmion tubes with homogeneous chirality. However, such possibly unique dynamics of the hybrid chiral skyrmion tubes have not been demonstrated, so it is unclear if these 3D topological spin structures could be apt for active elements in spintronic applications.

Here, we demonstrate unconventional non-reciprocal transport phenomena for 3D hybrid chiral skyrmion tubes, revealing, in particular, a current-induced non-reciprocal skyrmion Hall effect (NSkHE) in the flow regime for synthetic antiferromagnetic (SyAFM) hybrid chiral skyrmion tubes stabilized in multilayer systems at room temperature. The SyAFM system consists of multiple FM layers that are antiferromagnetically coupled via ultrathin non-magnetic spacers. This enables us to tailor the degree of non-reciprocity by controlling the long-range dipolar interactions as well as the interfacial and interlayer exchange interactions. The dynamics that exhibits a NSkHE does not yield significant differences in the velocity, and thus it is qualitatively different from the previously observed non-reciprocal dynamics of conventional 2D skyrmions with extrinsic in-plane symmetry breaking, e.g., from asymmetric confinements/pinning[36], or exchange bias[37], indicating a unique and intrinsic nature reflecting complex 3D topological spin textures. To understand the possible origin of the effect, we carry out micromagnetic modeling and find that hybrid chiral skyrmion tubes experience an asymmetric spin-orbit torque (SOT) contribution for the two current polarities due to helicity changes during their motion. This then induces the non-reciprocity observed in the skyrmion Hall effect. Our findings shed light on the internal degrees of freedom of skyrmion tubes, which so far have often been ignored, but are demonstrated to be key for cases such as the strong excitation regime provided by the current densities studied in this work[20]. Thus, this work provides crucial insights into the previously unexplored dynamics of 3D topological spin textures, and the complex dynamics observed could be a major asset for unconventional skyrmion-based technologies.

## Results

### Experimental setup and magnetic properties

We have prepared two synthetic antiferromagnets with different degrees of magnetic compensation to identify the repercussions of long-range dipole interaction and DMI on the dynamics of the skyrmion tubes. The stack structures consist of Ta(5.00)/[Pt(1.00)/ Ir(0.40)/$FM_{B1}$/Pt(1.00)/Ir(0.40)/$FM_{T1}$]$_{\times15}$/Pt(1.00) (numbers in nm) (ST1, hereafter), and Ta(5.00)/[Pt(1.00)/Ir(0.40)/$FM_{B2}$/Pt(1.00)/ Ir(0.40)/$FM_{T2}$]$_{\times25}$/Pt(1.00) (ST2, hereafter), respectively, deposited on 100 nm-thick silicon nitride membrane substrates (see Table 1 for the detail). The $FM_B$ and $FM_T$ layers are addressed as a bottom (Co-rich) FM and top (Fe-rich) FM layer, respectively, and they are antiferromagnetically coupled via the Pt/Ir interlayers. As shown in Fig. 1a, the compensation ratio of the magnetic moments, $m_C = 1 − |\mathbf{m}_T + \mathbf{m}_B|/ (|\mathbf{m}_T| + |\mathbf{m}_B|)$, was determined to be approximately 50% (ST1) and 80% (ST2) using magnetometry where the $\mathbf{m}_T$ and $\mathbf{m}_B$ denote the magnetic moments of $FM_T$ multilayers and $FM_B$ multilayers, respectively. The local SyAFM configuration, formed by a pair of adjacent CoB and CoFeB layers separated by the metallic spacer (see Table 1), is repeated 15 times, respectively 25 times, to induce the skyrmion tube structure. Our magnetization measurement suggests that no local breakdown of the antiferromagnetic (AFM) coupling between the magnetic domains occurs[42] in these stacks due to the sufficiently large interlayer exchange[18] resulting from the presence of Ir[43].

To directly demonstrate the interlayer AFM coupling that leads to fully coupled skyrmions in the layers before and after pulse injection (see "Methods"), we employed an element-specific detection technique: scanning transmission X-ray microscopy (STXM) with magnetic circular dichroism as the contrast mechanism, as illustrated in Fig. 1b. Owing to the characteristics of X-rays, the absorption and magnetic response of each element becomes energy-dependent, clearly indicating the presence of antiferromagnetically coupled skyrmions in our systems, see Fig. 1c, as expected from the $M$-$H_z$ curve. Since STXM is a transmission technique, note that the magnetic contrast is averaged across all layers containing the selected element. This confirms the presence of laterally and uniformly coupled skyrmions throughout the entire stack, which are described as skyrmion tubes. Figure 1c presents clear AFM skyrmion tubes using the $L_3$-edges of Fe (708.1 eV) and Co (778.6 eV) for stack ST2. Additionally, we confirmed that all the skyrmion tubes retain their AFM coupling after multiple current pulses, which indicates that all the skyrmions in the layers are fully coupled before and after their current-induced motion (see Supplementary Note 1). Furthermore, all skyrmion tubes exhibit no discernible multilevel gray XMCD contrast, confirming that inhomogeneous skyrmionic textures such as skyrmionic cocoons or bobber-style terminated skyrmions are absent owing to strong interlayer exchange coupling.

It is well known that the skyrmion size affects the quantitative evolution of the Hall angle in the skyrmion Hall effect[44–47]. Hence, by applying out-of-plane magnetic fields, we adjusted the skyrmion size to a fixed diameter of ~100 nm for each sample. In the next section, we systematically investigate the current-induced dynamics of the SyAFM skyrmion tubes.

### Current-induced dynamics of SyAFM skyrmion tubes

We first nucleate the skyrmion tubes by using a relatively high current density ($J \geq 15 \times 10^{11}$ Am$^{-2}$) under selected magnetic fields[48] and

---

## Table 1 | Building blocks of the SyAFM multilayer

| | FM$_B$, Bottom Co-rich FM layer | FM$_T$, Top Fe-rich FM layer | Repetitions |
|---|---|---|---|
| ST1 ($m_C$ = 50%) | FM$_{B1}$: [Co$_{0.8}$B$_{0.2}$(0.23)/Ta(0.06)]$_{\times3}$ /Co$_{0.8}$B$_{0.2}$(0.23) | FM$_{T1}$: Co$_{0.6}$Fe$_{0.2}$B$_{0.2}$(0.50)/ Co$_{0.2}$Fe$_{0.6}$B$_{0.2}$(0.40) | 15 |
| ST2 ($m_C$ = 80%) | FM$_{B2}$: Co$_{0.8}$B$_{0.2}$(0.35)/ [Ta(0.06)/Co$_{0.8}$B$_{0.2}$(0.4)]$_{\times2}$ | FM$_{T2}$: Co$_{0.6}$Fe$_{0.2}$B$_{0.2}$(0.50)/ Co$_{0.2}$Fe$_{0.6}$B$_{0.2}$(0.40) | 25 |

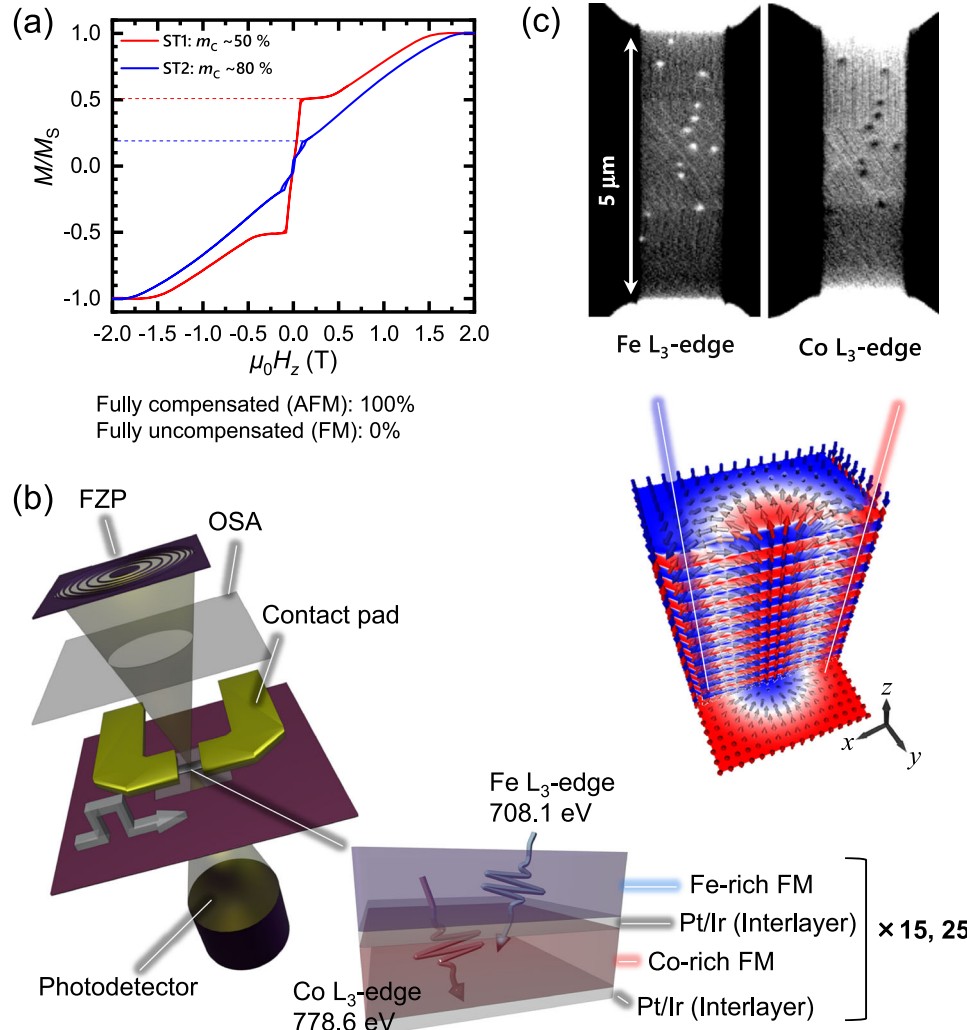

**Fig. 1 | Experimental setup, magnetization measurements, and element-specific detection of SyAFM skyrmion tubes. a** Normalized out-of-plane $M$-$H_Z$ curves for ST1 ($m_C \approx 50\%$) and ST2 ($m_C \approx 80\%$). **b** A schematic of the scanning transmission X-ray microscopy (STXM) setup. FZP Fresnel zone plate, OSA order sorting aperture, FM ferromagnet. **c** Direct observation of SyAFM coupling of skyrmion tubes under perpendicular magnetic field $\mu_0 H_Z = 130$ mT at room temperature, and schematic of the spin structure of the SyAFM skyrmion tube.

measured the average skyrmion velocity and average skyrmion Hall angle for stacks of different compensation. The fact that the SyAFM skyrmion tube moves along the technical current direction indicates that the spin-orbit torque (SOT) is the primary driving source for the current-induced motion[12,15,16,22,46,47]. To uncover the intrinsic dynamics, we track the trajectories of multiple SyAFM skyrmion tubes and link the trajectories using the Python module TrackPy (see Supplementary Movie 1), and then we take full statistics with respect to the individual skyrmion velocity, $v$, as can be seen in Fig. 2a, b. Figure 2c shows the current-density dependence of the average skyrmion velocity, $v_{ave}$, defined by the Gaussian peak of the statistics for each stack. We clearly find that the skyrmion tubes are more efficiently driven for ST2 due to the higher $m_C$, which is consistent with previous work[16]. Note that $v_{ave}$ is almost the same between the positive and negative pulses. Moreover, in Supplementary Movie 1, the SyAFM skyrmion tubes exhibit unambiguous diagonal motion, demonstrating a finite skyrmion Hall effect owing to the uncompensated magnetic moments between the top and bottom FM layers. We note that the absolute value of the skyrmion Hall angle depends on the size of the skyrmions and their domain wall width[49]: the skyrmion Hall angle inversely scales with the radius of the skyrmion for a given skyrmion domain wall width. To scrutinize the current-induced

SyAFM skyrmion tube dynamics, we proceed by thoroughly investigating the skyrmion Hall effect.

## Non-reciprocity in the skyrmion Hall effect for the low-compensation SyAFM skyrmion tubes

For a quantitative evaluation of the skyrmion Hall effect, we employed the same protocol as for the skyrmion velocity, where the averaged skyrmion Hall angle, $\theta_{ave}$, is extracted from the peak of the Gaussian fitting. Figure 3a presents the statistics of the skyrmion Hall angle for the skyrmion ensembles in the stack ST1 ($m_C = 50\%$). We find that the values of $\theta_{ave}$ are reciprocal for both current pulse polarities at $v_{ave} = 10$ m s$^{-1}$, which corresponds to the creep/depinning regime (see Fig. 2c). Surprisingly, we find an intriguing behavior for the higher velocity in the flow regime of $v_{ave} = 25$ m s$^{-1}$, where the degeneracy of two Gaussian peaks regarding the current polarity is distinctly lifted, indicating the presence of a non-reciprocal skyrmion Hall effect in the flow regime (see also Supplementary Movies 1 and 2). To ascertain the physical origin of the non-reciprocity in the Hall angle, we move on to measure the average velocity dependence of $\theta_{ave}$. As depicted in Fig. 3b, the NSkHE is found to vanish in the creep/depinning regime, as anticipated from Fig. 3a. This implies that the intrinsic mechanism dominates over the pinning effects[37] and is essential for the observed

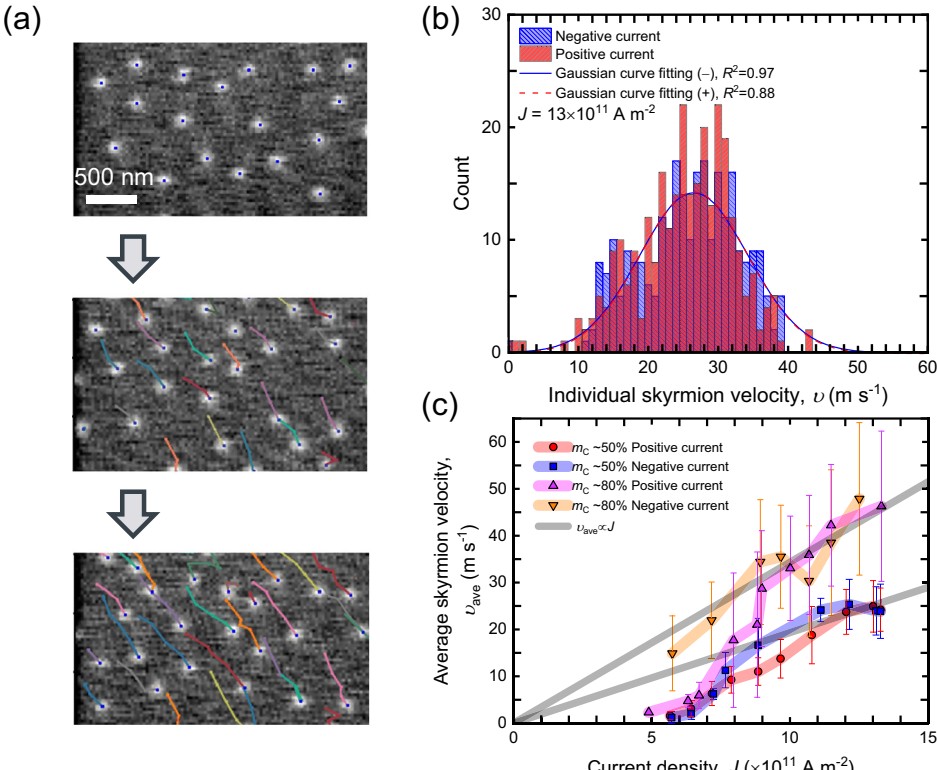

**Fig. 2 | Current-induced motion of SyAFM skyrmion tubes. a** Representative tracked trajectories of current-induced motion of SyAFM skyrmion tubes. **b** Statistical analysis of the individual SyAFM skyrmion tube velocity for the stack ST1 ($m_C = 50\%$) under $\mu_0 H_Z = 70$ mT at $J = 13 \times 10^{11}$ A m$^{-2}$. Red and blue colors correspond to the positive and negative current polarity, respectively. The curves denote the Gaussian curve fitting. **c** The current-density dependence of the average velocity, $v_{ave}$, is defined

by the Gaussian peak positions with ±1σ bars of the curve fitting in Fig. 2b. Red circles, blue rectangles, magenta triangles, and orange inverse triangles represent positive current polarity (ST1 stack), negative polarity (ST1 stack), positive polarity (ST2 stack), and negative polarity (ST2 stack), respectively. In line with literature, we use the common definition of the flow regime as the region in which the average skyrmion velocity exhibits a linear dependence on the applied current density.

strong NSkHE. Intriguingly, the NSkHE becomes negligible for the high-compensation stack (ST2, $m_C = 80\%$), as it is evident from Fig. 3b. This dependence of the NSkHE on the degree of magnetic compensation unveils that the long-range dipole interaction, which is significant only for ST1, is one of the critical factors for tailoring the non-reciprocity.

## Micromagnetic simulations and discussion

Next, we analyze the dynamics of the SyAFM skyrmion tubes, where reversing the direction of the applied current results in a different skyrmion Hall angle, indicating strongly non-reciprocal behavior. In ferromagnetic (FM) multilayers, long-range magnetic dipolar interactions facilitate the formation of hybrid chiral skyrmion tubes, which consist of Bloch-type skyrmions at the center of the tubes and Néel-type skyrmions of opposite chiralities at the surfaces[40,41]. These hybrid skyrmion tubes can exhibit a multifaceted response to SOTs[50,51], leading to a rich tapestry of dynamic effects. However, such exotic hybrid chiral skyrmion dynamics have yet to be demonstrated experimentally. In particular, long-range dipolar interactions are predominant in the SyAFM system with substantial uncompensated moments, potentially acting as a stabilizing agent for the antiferromagnetically coupled hybrid skyrmion tube: the "hybridness" of the skyrmion tubes, defined as the ratio of Néel to Bloch components, can be modulated by adjusting the magnetic compensation ratio and the number of repetitions within the SyAFM multilayers, thus offering a means to control the dynamics of these skyrmions. To check this conjecture, we next employ micromagnetic simulations using the MuMax3 software[52], with parameters

extracted from experimental data as detailed in the "Methods" section.

The spin structure of the skyrmion tube obtained from relaxation at zero field using micromagnetic simulation provides further insights into the effect of magnetic compensation on the equilibrium structure. These simulations reveal that the skyrmion helicity varies across different layers due to net stray fields arising from the net magnetization in the multilayer system. Figure 4a illustrates the 2D spin profiles of individual layers of the SyAFM multilayer, indicating the orientation of magnetization: red for magnetization along the +$x$-axis and blue for the −$x$-axis. For the 80% compensated stack (ST2), the spin profiles show homochiral Néel-type skyrmion tubes throughout the multilayer stack, with each layer exhibiting a helicity shift of π due to the AFM coupling. In contrast, at 50% compensation, a Néel-type domain wall characterizes one end of the skyrmion tube with helicity ($\eta$) being zero, while the helicity varies continuously through successive layers, maintaining AFM alignment at their cores. The simulation also indicates that the hybrid character should have a subtle but finite Bloch component, which is evidenced by Lorentz transmission microscope imaging (TEM) (details, see Supplementary Note 2).

Figure 4b elucidates the velocity of hybrid chiral skyrmion tubes as a function of current density obtained via micromagnetic simulation (see "Methods" for details). Our results reveal that the velocity of the hybrid skyrmion tubes exhibits a near-linear response to the current density, paralleling the behavior in the experimental results, as depicted in Fig. 2c. Most importantly, Fig. 4c illustrates a distinct transition in skyrmion dynamics: within region 1, the skyrmion Hall angle remains reciprocal, indicating an absence of directional bias.

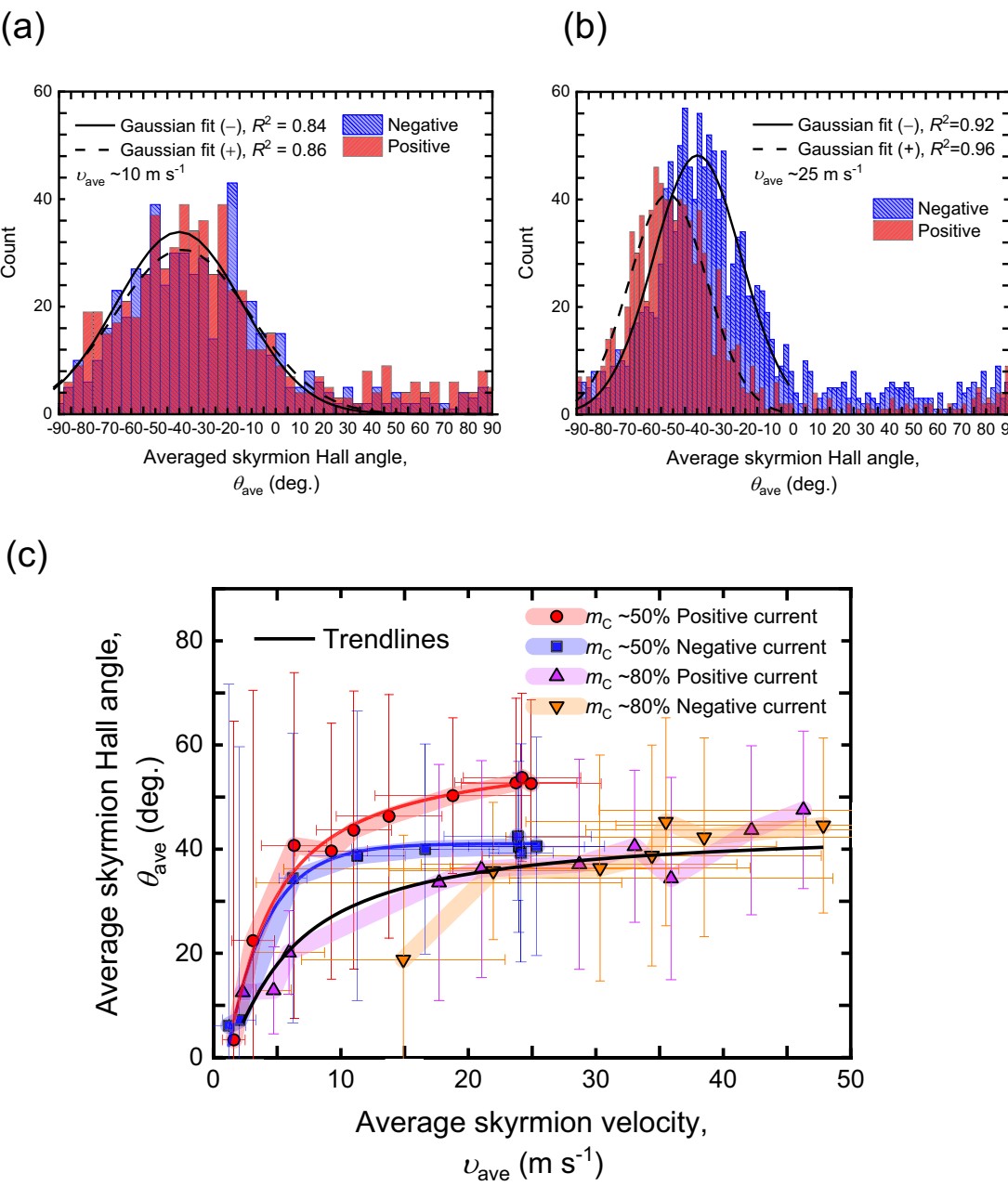

**Fig. 3 | Dependence of the non-reciprocal skyrmion Hall effect on the degree of magnetic compensation. a** Statistical analysis of the individual skyrmion Hall angle for the stack ST1 ($m_C = 50\%$) under $\mu_0 H_Z = 70$ mT at $v_{ave} = 10$ m s$^{-1}$ and **b** 25 m s$^{-2}$. **c** The average velocity dependence of $\theta_{ave}$ with $\pm 1\sigma$ bars of the curve fitting extracted from the Gaussian peaks in Fig. 3a. Red circles, blue rectangles, magenta triangles, and orange inverse triangles represent positive current polarity (ST1 stack), negative polarity (ST1 stack), positive polarity (ST2 stack), and negative polarity (ST2 stack), respectively.

However, beyond a critical velocity threshold of ~30 m s$^{-1}$, defining the onset of region 2, there is a significant increase in the skyrmion Hall angle, with different amounts for opposite directional currents, resulting in a prominent non-reciprocal behavior.

Non-reciprocal dynamics have been reported for ferrimagnetic insulators with effective in-plane exchange bias[37], where the shape distortion in the skyrmion configuration leads to a finite difference for the off-diagonal elements in the dissipation tensor due to asymmetric pinning effects[37,53]. The field-like torques (FLTs) also induce skyrmion shape distortions, possibly leading to an asymmetric skyrmion Hall angle[46]. Although the NSkHE could be induced via these scenarios, this is not the case for our stacks: First of all, when asymmetric pinning causes non-reciprocity, the NSkHE should emerge with a significant

velocity difference even in the creep and depinning regimes via the shape or velocity-dependent skyrmion Hall effect[22,44–47] as previously observed[37,53]. Secondly, our spin-torque ferromagnetic resonance (ST-FMR) measurements reveal comparably small FLT components compared to the damping-like torque (DLT) (see Supplementary Note 3). To substantiate the above discussion, we have investigated the shape distortions by computing the dissipation tensor, $D_{\mu\gamma} = \int_S d^2\mathbf{r} \left( \partial_\mu \mathbf{m}_i \bullet \partial_\gamma \mathbf{m}_i \right)$, for the SyAFM hybrid chiral skyrmion in the aforementioned regions (see Supplementary Note 4). Figure 5a, b illustrates the differences $\widetilde{D}_{\mu\gamma} = D_{\mu\gamma}[+J] - D_{\mu\gamma}[-J]$ of the lateral average of the dissipation tensor for two antiparallel directions of the applied current. At the skyrmion velocity of $v = 11$ m s$^{-1}$ (region 1), the tensor elements, $\widetilde{D}_{xx}$, $\widetilde{D}_{xy}$, $\widetilde{D}_{yx}$, and $\widetilde{D}_{yy}$, show no deviation, indicating

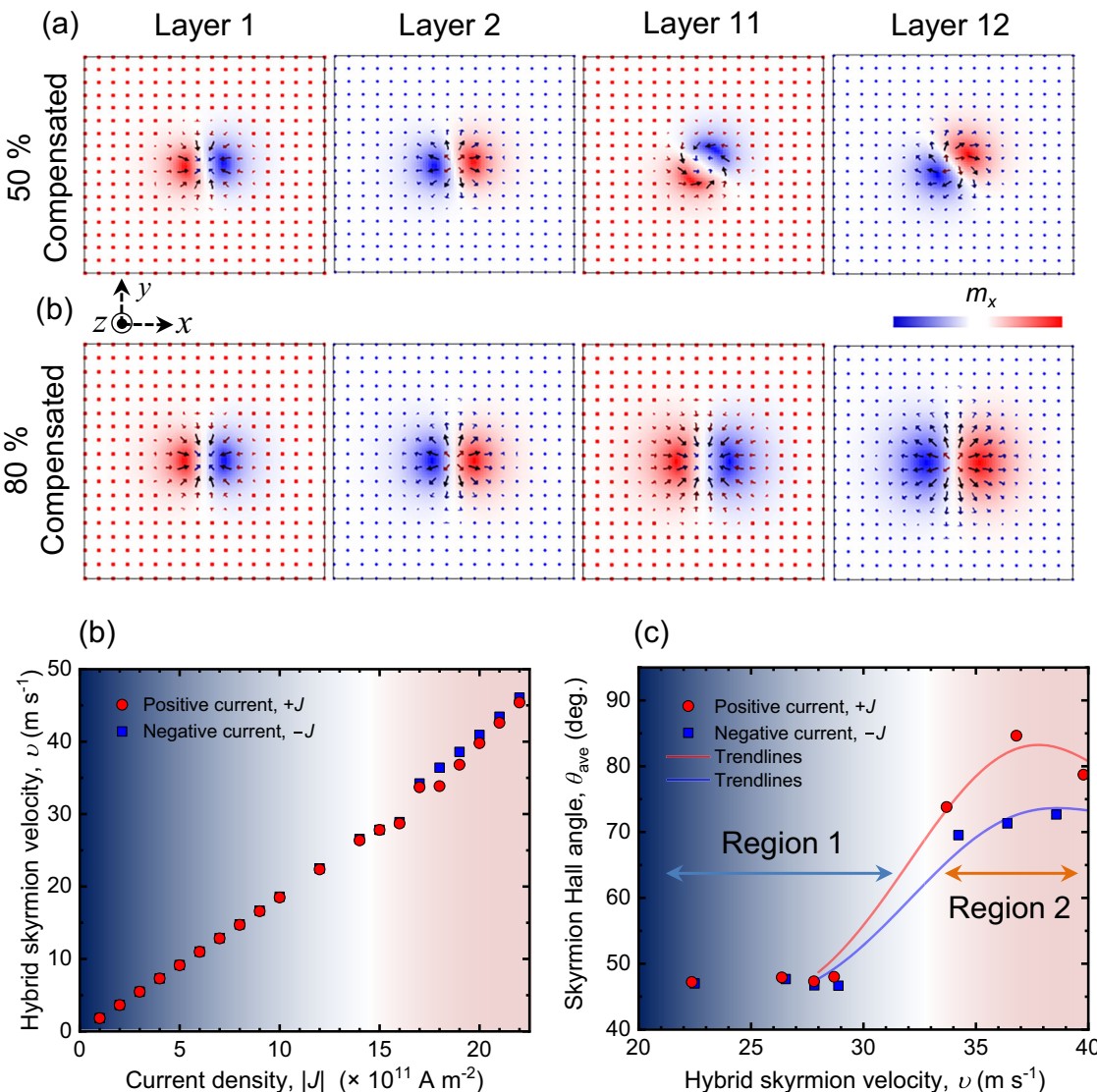

**Fig. 4 | Micromagnetic simulations of antiferromagnetic hybrid chiral skyrmion tubes.** **a** Two-dimensional (2D) profiles of magnetic skyrmions in layers 1, 2, 11, and 12 for SyAFM systems with $m_C = 50\%$ and $m_C = 80\%$ compensation, respectively. Arrow directions indicate the magnetization orientation within each layer, while color variations denote the magnetization direction along the $x$-axis. For the $m_C = 80\%$ compensated SyAFM system, the skyrmion helicity remains constant across all layers. In contrast, for the $m_C = 50\%$ compensated SyAFM, the skyrmion helicity transitions from Néel to a hybrid type. **b** The velocity of hybrid skyrmions in the $m_C = 50\%$ compensated SyAFM system, depicted as a function of the current density for two antiparallel in-plane ($x$-axis) current directions. **c** Variation of the skyrmion Hall angle for hybrid skyrmion tubes, as a function of skyrmion velocity for two antiparallel in-plane ($x$-axis) current directions, showcasing the non-reciprocal response of skyrmions to applied electrical currents at higher current density.

that the skyrmion remains circular during its motion. At a higher velocity of $v = 34$ m s$^{-1}$ (region 2), the tensor elements change when the direction of the applied current is reversed. However, these imperceptible deviations are insufficient to cause the observed strong NSkHE. This certainly shows that shape distortions do not predominantly contribute to the non-reciprocity of the skyrmion Hall angle, which necessitates a different mechanism to describe our experimental observations.

Here, we unravel the possible physical origin of the NSkHE for the SyAFM hybrid chiral skyrmion tube by computing the SOT efficiency tensor, $I_{\mu\gamma} = \int_S d^2\mathbf{r}\,[\partial_\mu\mathbf{m}_i\mathbf{m}_i]_\gamma$ (see Supplementary Note 4) in both regions of the skyrmion tube motion. Figure 5c, d shows the difference, $\tilde{I}_{\mu\gamma} = I_{\mu\gamma}[+J] - I_{\mu\gamma}[-J]$, of the lateral average of the SOT tensor for two opposite directions of the applied currents: in region 1 (Fig. 5c), the $\tilde{I}_{\mu\gamma}$ is negligibly small and comparable to the $\tilde{D}_{\mu\gamma}$. However, in region 2 (Fig. 5d), it is at least two orders of magnitude higher than

those of $\tilde{D}_{\mu\gamma}$, therefore indicating that this is the main contribution leading to the non-reciprocity of the skyrmion Hall angle.

To clarify the mechanism for such a change in the SOT tensors, we further compute the variation of the average helicity of the hybrid skyrmion tubes during the current-induced dynamics. In region 1, the skyrmions retain their helicity during motion, which is unambiguously seen in Fig. 5e (also see Supplementary Movie 3 and 4), and thus the difference in the average of the $I_{\mu\gamma}$ elements is negligible as expected. However, in region 2, layers with Néel-type characters change their helicity and acquire more of a Bloch-type character (see Supplementary Movie 5 and 6), causing the non-reciprocity in the skyrmion Hall angle as shown in Fig. 5f. The dynamical behavior of the skyrmion helicity is a key feature that warrants a detailed discussion. Our results indicate that the helicity transition from Néel to the Bloch type (namely helicity ranging from zero to π/2) occurs across the $z$-direction in region 2, where the current density is higher. This transition is more

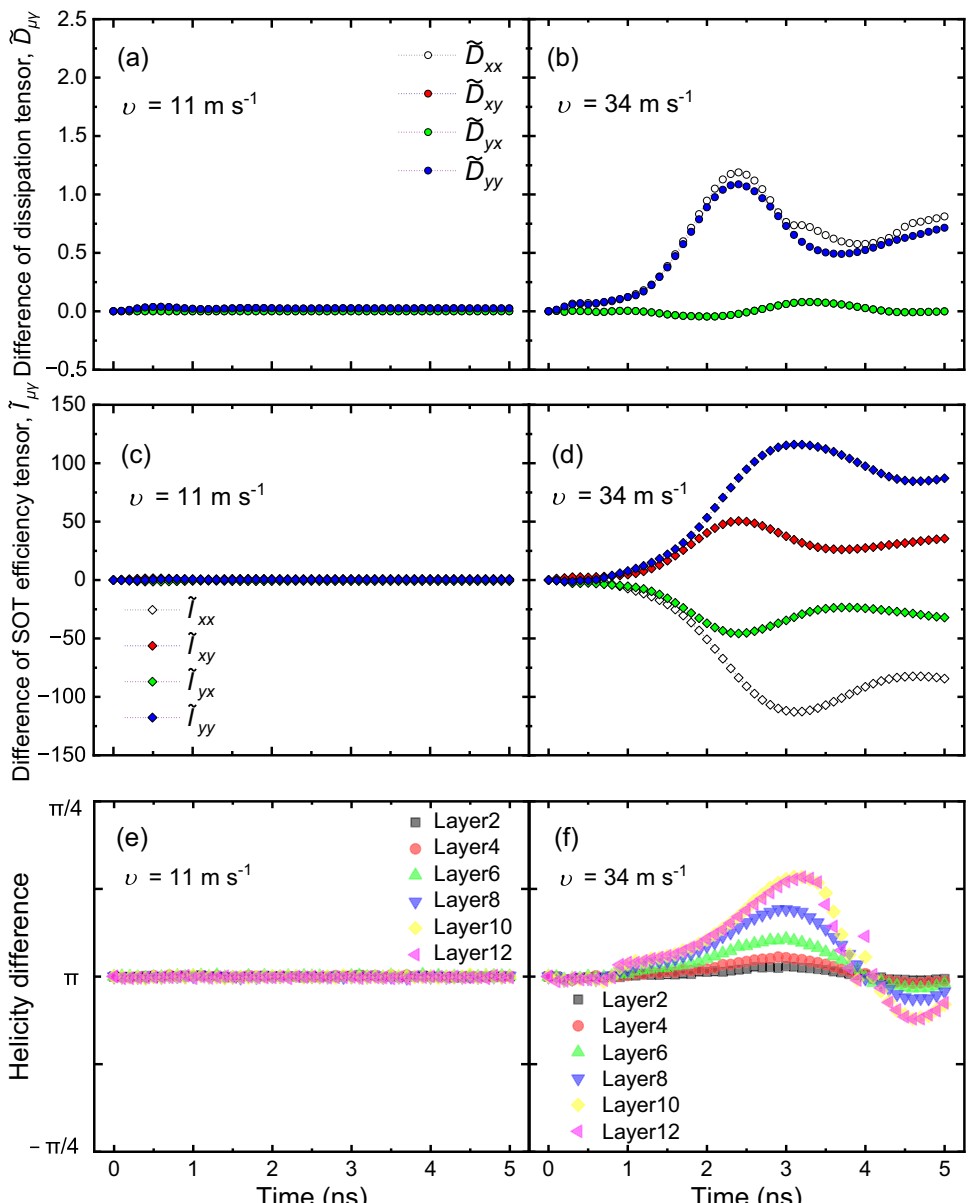

**Fig. 5 | Evolution of both dissipation and SOT efficiency tensors ($D_{\mu\gamma}$ and $I_{\mu\gamma}$, respectively) in two different regions of skyrmion dynamics.** Difference of the dissipative tensors (layer-averaged) at the skyrmion tube velocity of **a** $\upsilon = 11\,\mathrm{m\,s^{-1}}$ and **b** $34\,\mathrm{m\,s^{-1}}$. Difference of the SOT efficiency tensors (layer-averaged) at the skyrmion tube velocity of **c** $\upsilon = 11\,\mathrm{m\,s^{-1}}$ and **d** $34\,\mathrm{m\,s^{-1}}$. The *xx*, *xy*, *yx*, and *yy* components correspond to white, red, green, and blue colors, respectively. Layer-dependent helicity difference between positive and negative pulses at the skyrmion tube velocity of **e** $\upsilon = 11\,\mathrm{m\,s^{-1}}$ and **f** $34\,\mathrm{m\,s^{-1}}$, illustrating the asymmetry that leads to non-reciprocity in the skyrmion Hall effect.

pronounced in magnetic layers where the skyrmion texture exhibits a Bloch character in equilibrium (see Supplementary Note 5). This non-uniform variation of the helicity across the layers is significant since the resultant oscillations in time in the values of the helicity cannot be treated as minor perturbations; on the contrary, they significantly alter the skyrmion dynamics, leading to the non-reciprocal Hall effect that we observe. Therefore, the dynamics of the hybrid-skyrmion helicity is a critical factor driving the observed non-reciprocity.

The transition in the helicity and its resulting impact on the skyrmion dynamics highlight the intricate interplay between magnetic interactions in SyAFM multilayers. This complexity is further compounded by dipolar interactions, that can be tuned by the level of compensation. Our findings show that these interactions are crucial for stabilizing the hybrid chiral skyrmion tubes and modulating their behavior under applied currents. Dipolar interactions, combined with

the uncompensated magnetic moments, create a unique environment where skyrmion dynamics can be finely tuned by adjusting the magnetic compensation ratio and the number of repetitions within the multilayers.

We note that while the parameters used in the micromagnetic simulations are obtained experimentally (see "Methods" section), there are limitations to the quantitative analysis. For instance, the phenomenological damping obtained from our experiments has multiple contributions, including inhomogeneous broadening and interfacial spin mixing. Additionally, we are unable to experimentally distinguish the contribution of the Oersted field from the intrinsic symmetry of the FLT. These key factors would affect the quantitative agreement with the experimental observation. Also, since our micromagnetic calculations are done at zero temperature, we cannot expect a full quantitative agreement. Therefore, our calculations are used to

provide the key qualitative explanations of the major features in the experimental observation. The unambiguous conclusion that we can draw is that the observed NSkHE without any extrinsic in-plane symmetry breaking highlights the critical role of the internal degree of freedom in the current-induced motion of the skyrmion tubes.

## Discussion

So far, the dynamics of topological spin structures in ferrimagnetic/AFM systems have been analyzed in terms of the angular momentum compensation, e.g., leading to a vanishing skyrmion Hall effect. The other key aspect of the antiferromagnetically coupled system is the magnetic compensation, whose role in the skyrmion dynamics has yet to be fully understood. Our results clearly show the role of magnetic compensation in the skyrmion tube dynamics in thin-film multilayer systems. The magnetic compensation level modulates the internal degree of freedom in skyrmions, and thus allows for triggering the NSkHE via the change in the SOT efficiency resulting from the variation of the helicity across the FM layers. This simultaneously provides an excellent tunability of the NSkHE. The non-reciprocal behavior observed in our experiments is a direct consequence of the complex interplay between magnetic dipolar interactions and the intrinsic properties of the skyrmion strings. As the observed intrinsic NSkHE does not show a significant velocity difference (see Fig. 2c), it is distinctly different from the previously reported non-reciprocal transport of 2D skyrmions driven by an additional extrinsic in-plane symmetry breaking in complex systems[36,37], and thus a unique nature of 3D topological quasi-particles. Such intrinsic NSkHE fully leveraging the helicity modulation enables us to devise simplified device configurations that are favorable for applications.

FM skyrmion tubes potentially show a similar behavior but this has not been demonstrated yet, as the Bloch chirality degeneracy is presumably not lifted in FM skyrmion tubes with only the interfacial DMI ($C_{nv}$) and dipolar coupling present. Recently, it has been reported that the interlayer DMI could lift the degeneracy and lead to a large asymmetry in the population of Bloch chiralities[54]. Indeed, our experimental observations in Fig. 3a present a similar statistical asymmetry as observed in the work mentioned above[54]. In this sense, magnetic systems with AFM coupling and controlled interlayer DMI, such as SyAFM multilayers, would be a promising platform to investigate the current-induced dynamics with internal degrees of freedom of 3D topological spin textures.

In summary, we have demonstrated the interlayer-exchange-interaction-mediated coherent antiferromagnetic coupling of 3D SyAFM skyrmion tubes, and their remarkable robustness against current excitations, as directly observed using element-specific STXM. Our comprehensive investigation into the dynamics of hybrid chiral skyrmion tubes in the SyAFM multilayer reveals a unique character based on 3D topological spin profiles: an intrinsic non-reciprocal skyrmion Hall effect, generated by dynamic variations of the helicity rather than shape distortions. With the excellent tunability by the magnetic compensation, this behavior opens avenues for further exploration into the dynamical properties of 3D hybrid skyrmion tubes and their potential for advanced applications in spintronic devices that require complex dynamics. Classical and unconventional computing, as well as energy harvesting possibly leveraging the non-reciprocal emergent electric fields generated by the nonlocal NSkHE would be promising directions. Hence, our findings pave the unexplored way for topological spin texture-based unconventional electronic technologies.

## Methods
### Film preparation and device fabrication
The SyAFM stack structures were deposited on 100-nm-thick $Si_3N_4$ membranes for the observation of the dynamics of SyAFM skyrmion tubes as well as on 0.5-mm-thick $Si_3N_4$ (100 nm)/Si substrates for the pre-characterization of magnetic properties using a Singulus Rotaris sputtering system with a base pressure lower than $3 \times 10^{-8}$ mbar. In this work, the thickness of the bottom $Co_{0.8}B_{0.2}$/Ta multilayer was controlled to tailor the magnetic compensation. To apply the current pulses, we patterned 5–15 μm-wide wires with contact pads made of Cr/Au by a conventional lift-off process using electron-beam lithography technique. The gap between the contact pads was roughly 1.5–4.5 μm, enabling us to obtain impedance matching, and thus reach current densities in the order of $10^{12}$ A m$^{-2}$ for narrow pulse widths (5–15 ns), which is large enough to drive the skyrmion tubes in the flow regime. For the ST-FMR measurements, additional 40-μm-long and 10-μm-wide wires in a coplanar waveguide layout were fabricated in the same process.

### Magnetic parameters
Magnetization (m-H) curves were recorded using a superconducting quantum interference device (SQUID), with the saturation magnetization ($M_S$) for the top and bottom layers measured at 1.57 T and 0.5 T, respectively. The effective magnetic anisotropy energy density ($K_{eff}$) was determined to be 0.11 MJ m$^{-3}$. Furthermore, the presence of a FM-AFM phase transition was confirmed, from which the magnetic compensation ratio $m_C$ was experimentally obtained. The magnitude of the interfacial DMI $D_i$ was estimated to be 0.4–0.5 mJ m$^{-1}$ based on our previous observations[18], which reproduces reasonably well the experimentally observed size of the SyAFM skyrmion tubes. Microwave-excited FMR and ST-FMR measurements were used for experimentally obtaining the interlayer exchange coupling fields $\mu_0 H_{ex} \sim 0.2$ T, the effective damping constant $\alpha = 0.1$, and the effective spin Hall angle $\theta_{SH} = 0.1$ in the SyAFM system[55,56] (see Supplementary Note 3 for details). These parameters are used in the micromagnetic modeling of the current-induced dynamics of the skyrmion tubes.

### Element-specific magnetic domain observation and current-induced dynamics
The element-specific magnetic domain imaging by scanning transmission X-ray microscopy and the current-induced motion of SyAFM skyrmion tubes were conducted at the MAXYMUS endstation of the BESSY II electron storage ring operated by the Helmholtz-Zentrum Berlin für Materialien und Energie and at the PolLux beamline of the Swiss Light Source. X-ray absorption spectra taken beforehand allow us to identify the corresponding X-ray energies of Fe L$_3$-edge and Co L$_3$-edge to be 708.1 eV and 778.6 eV, respectively. The unambiguous AFM coupling of the skyrmion tubes was confirmed with these energies, as shown in Fig. 1c. Moreover, it was confirmed that the AFM coupling was retained after current pulsing in the full range of current densities explored in this work (see Supplementary Note 1).

Skyrmion tubes in our system nucleate spontaneously near the spin reorientation transition (SRT), where the effective anisotropy approaches zero and may become negative. Under these conditions, the uniform magnetic state becomes unstable, and spin textures including chiral skyrmions are favored. In this regime, the interfacial Dzyaloshinskii-Moriya interaction and long-range dipolar interactions outweigh the exchange and anisotropy energies, driving the formation of a multidomain ground state that can include stripe domains and skyrmions. By adjusting the thickness of each FM layer in our SyAFM multilayers, we tune the system into this SRT regime and thus promote spontaneous multidomain formation. Skyrmions are selectively stabilized by applying an out-of-plane magnetic field, which penalizes antiparallel domain configurations and favors isolated circular textures. As the energy landscape is nearly degenerate, with circular skyrmions and stripe domains coexisting, external stimuli are required to overcome the nucleation barrier. To initialize well-defined skyrmion states across different compensation levels, we employ two methods: (1) out-of-plane magnetic field cycling and (2) current-induced nucleation.

After the current-induced nucleation of the skyrmion tubes under specific perpendicular magnetic fields ($\mu_0 H_z = 70–130$ mT), the size of individual skyrmion tubes was adjusted to 100 nm within 15% deviation by applying perpendicular magnetic fields of 70 mT (for the stack ST1) and 130 mT (for the stack ST2). To drive the SyAFM skyrmion tubes, current pulses with a width of 5–15 ns were injected through the Cr/Au contact pads. The Python module Trackpy[57] was employed to detect the skyrmion tubes and to track and link their trajectories during the motion. Mimicking our previous works[26,28,58], the appropriate conditions for tracking and linking were successfully obtained, which allows for the statistical evaluation of the skyrmion tube ensembles, as shown in Figs. 2 and 3 in the main text. In the statistical dataset corresponding to Fig. 3a, each bin on the horizontal axis spans 4°, reflecting the resolution of this dataset. Note that the obtained maximum of the skyrmion-tube velocity was limited by the current-induced nucleation of SyAFM skyrmion tubes at relatively higher current densities ($J > 13 \times 10^{12}$ A m$^{-2}$), where tracking and linking were unfeasible anymore owing to multiple nucleation events.

## Micromagnetic simulations

To investigate the non-reciprocal behavior of the skyrmion motion, we performed micromagnetic simulations of the SyAFM skyrmion at various current strengths using the MuMax3 software. The simulation setup consists of a multilayer stack structure (12 ferromagnetic layers) where layers with opposite magnetizations were coupled by the interlayer exchange interaction. The multilayer stack geometry considered had each layer with a lateral size of $256 \times 256$ nm$^2$ and a thickness of 1 nm. The system was discretized with a mesh size of $(1 \times 1 \times 1)$ nm$^3$, and periodic boundary conditions were imposed along the $x$ and $y$ directions with a period of 8 repetitions. The dipole-dipole interactions were included implicitly in the calculations to accurately account for the inter-layer dipolar effects. The material parameters used in the calculations in Fig. 5 were: exchange constant $A = 10$ pJ m$^{-1}$, interfacial DMI strength $D_{top} = 0.45$ mJ m$^{-2}$, $D_{bottom} = 0.45$ mJ m$^{-2}$, and Gilbert damping $\alpha = 0.1$. The interlayer exchange coupling strength was $\lambda = 0.4$ mJ m$^{-2}$, corresponding to the value obtained from the magnetometry and FMR measurements. To match the compensation ratio of magnetic moments determined to be approximately 50%, the magnetic moments of CoFeB multilayers and CoB multilayers were set to $M_{S\_top} = 1.56$ T and $M_{S\_bottom} = 0.5$ T, respectively, which are the saturation magnetization of top and bottom layer, respectively. Similarly, the uniaxial anisotropies of the two layers were set to $K_{U\_top} = 1$ MJ m$^{-3}$ and $K_{U\_bottom} = 1.1 \times 10^5$ J m$^{-3}$, respectively. Note that in order to keep computational cost manageable, we used twelve repetitions of the multilayer stack, which was the maximum number allowing for reasonable convergence. To clarify the underlying physical picture, we performed a systematic and detailed investigation of the magnetic compensation dependence, as presented in Supplementary Note 6.

## Data availability

The source data supporting the findings of this study are provided within the main text and the Supplementary Information files. The statistical datasets underlying Fig. 2 and Fig. 3 are available at Zenodo: https://doi.org/10.5281/zenodo.16797254.

## Code availability

The computer codes used for data analysis are available upon reasonable request from the corresponding author.

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

## Acknowledgements

This work was partly supported by JSPS Kakenhi (Grant Nos. 23K13655, 24H00039, 24H00409, 25K01645) as well as the Center for Science and Innovation in Spintronics at Tohoku University (Cooperative Research Project). We thank Helmholtz-Zentrum Berlin für Materialien und Energie for the allocation of synchrotron radiation beamtime. The PolLux end station was financed by the German Ministerium für Bildung und Forschung (BMBF) through ErUM-Pro contracts 05K16WED, 05K19WE2, and 05K22WE2. This work has received funding from the European Union's Horizon 2020 research and innovation program under the Marie Skłodowska-Curie Grant Agreement No. 860060 "Magnetism and the effect of Electric Field" (MagnEFi), as well as from Synergy Grant No. 856538, project "3D-MAGiC," the Horizon Europe Project No. 101070290 (NIMFEIA) and Horizon Europe Programme Horizon 1.2 under the Marie Skłodowska-Curie Actions (MSCA), Grant agreement No.101119608 (TOPOCOM). It has also been supported by the Deutsche Forschungsgemeinschaft (DFG, German Research Foundation) – SPP 2137 Skyrmionics (project 462597720), TRR 173 – 268565370 (project A01, B02 and A03), TRR 288 – 422213477 (project A09), project 445976410, project 448880005, and the Dynamics and Topology Center TopDyn funded by the State of Rhineland Palatinate. This work was also partly supported by the Norwegian Research Council through its Center of Excellence, Project Number 262633, "QuSpin".

## Author contributions

T.Dohi and M.K. initiated, designed, and supervised the project. T.Dohi, M.B., F.K., and M.A.S. designed and prepared the stack structure as well as fabricated it into the μm wires. T.Dohi, M.B., F.K., M.A.S., and D.M.T. pre-characterized the stacks using the superconducting quantum interference device (SQUID) and measured the current-induced skyrmion tube dynamics using the STXM with technical support from S.W., M.W., S.F., and J.R.; T.Dohi analyzed the experimental data for the current-induced dynamics. V.K.B. and M.B. performed the micromagnetic simulations with input from R.Z.; V.K.B. and R.Z. did the numerical analysis and analytical modeling. A.S. measured the ferromagnetic resonance (FMR) for evaluating the damping constant and the spin Hall angle. T.Denneulin conducted the magnetic domain imaging using the Lorentz TEM with input from R.E.D.; T.Dohi, M.B., V.K.B., and R.Z. drafted the manuscript with the help of R.F., J.S., and M.K. All authors discussed the results and commented on the manuscript.

 

## Funding

## Competing interests

The authors declare no competing interests.
