## [Peer Review File · Nature Communications]

Observation of a non-reciprocal skyrmion Hall effect of hybrid chiral skyrmion tubes in synthetic antiferromagnetic multilayers

Corresponding Author: Professor Mathias Kläui

Version 0:

Reviewer comments:

Reviewer #1

(Remarks to the Author)

In the work by Dohi et al., the authors have investigated the non-reciprocal transport phenomena of the 3D hybrid chiral skyrmion tubes in synthetic antiferromagnetic (SyAFM) multilayers. The authors attributed this non-reciprocal behavior to the asymmetric spin-orbit torque (SOT) effects that is associated with the dynamic changes of the skyrmion tube during its motion. Specifically, the helicity of the skyrmion evolves dynamically during motion, which results in the non-reciprocal behavior, rather than being caused by shape distortion. The experiments and simulations seem systematic. However, the current version of the manuscript is not suitable for publication in Nature Communications. I would encourage authors to revise the manuscript based on the following comments and extend necessary discussions and experiments. I would be happy to take a second look of the revised manuscript.

- (1) On Page 4, the authors mentioned that the observed non-reciprocal skyrmion Hall effect is qualitatively different from the previously observed non-reciprocal dynamics of conventional 2D skyrmions with extrinsic in-plane symmetry breaking... However, no relevant references were inserted at this point, which makes this referee hard to follow.
- (2) On Page 5, what is the unit of thickness inside the parentheses in the stack structures of ST1 and ST2?
- (3) On Page 6, the authors mentioned that the magnetic contrast in the STXM results represents the averaged response of all layers. How did the authors confirm the presence of skyrmions in each individual layer? How authors can rule out the existence of layer dependent skyrmionic textures, which is typical in skyrmion hosting multilayers?
- (4) The authors should include some images of skyrmion motion in the main text to help readers better visualize and understand the experimental phenomena.
- (5) In Figure 3b, when the average skyrmion velocity reaches 10 m/s, the skyrmion Hall angles under positive and negative currents show a significant difference, which does not consistent with Figure 3a. The authors should provide an explanation.
- (6) The authors should provide the definitions of “creep/depinning regime” and “flow regime” in the main text. Please also clarify how authors identify these regimes.
- (7) The authors proposed that the non-reciprocal skyrmion Hall effect is closely related to the degree of compensation. Could the authors provide a micromagnetic simulation result showing how the skyrmion Hall effect changes from a partly compensated (FIM) state, through various degrees of compensation, to a fully compensated (AFM) state?
- (8) The authors mention that the non-reciprocal skyrmion Hall effect is also closely related to the “hybridness” of skyrmion tubes. If authors wanted to bridge this hybridness with hybrid skyrmion in multilayers, I would like to see the evidence of hybrid skyrmions. In addition, authors should explain how to precisely control the “hybridness” of skyrmion tubes experimentally?
- (9) In the magnetic multilayers with low values of DMI, due to the influence of dipolar interactions, different chiral spin structures may appear in the top and bottom layers. Could the authors add the simulation results for the top layer (layer 15 for ST1 and layer 25 for ST2) in Figure 4a?
- (10) The labels in Figure 4 are incorrect. The authors should carefully check the manuscript to avoid discrepancies between the main text and the figures.

Reviewer #2

(Remarks to the Author)

This manuscript presents the observation of a non-reciprocal skyrmion Hall effect (NSkHE) in hybrid chiral skyrmion tubes within synthetic antiferromagnetic (SyAFM) multilayers. By combining experimental characterization with micromagnetic simulations, the study highlights the critical role of dynamic helicity oscillations in this phenomenon. The work is of significant scientific merit, offering valuable insights into the dynamics of 3D topological spin textures and their potential applications in spintronics. However, the following issues need to be addressed or clarified before the manuscript can be considered for acceptance:

1. Figures 2a and 3a: Include goodness-of-fit metrics (e.g., R^2 values) for the Gaussian fits. Additionally, provide data from repeated experiments to confirm the reproducibility of the results.
2. Skyrmion Diameter Stabilization: Clarify the mechanism and range of stabilization for skyrmion diameters (~100 nm) under out-of-plane fields. Discuss how variations in skyrmion size ($\pm 15\%$) may influence the NSkHE and its dynamics.
3. Experimental Evidence of Helicity Oscillations: While micromagnetic simulations suggest helicity oscillations, direct experimental evidence (e.g., time-resolved, layer-specific magnetic imaging via STXM or Lorentz microscopy) is absent. If feasible, include such data; otherwise, elaborate on how the simulations align with experimental observations to strengthen the interpretation.
4. Magnetic Compensation: The NSkHE is observed exclusively at 50% magnetic compensation. Discuss whether this phenomenon could extend to other compensation levels or material systems, such as ferromagnetic multilayers.
5. Simulation Parameters: Explicitly state whether the damping constant ($\alpha = 0.1$) and interfacial DMI strength ($D = 0.45$ mJ/m²) used in the simulations correspond to experimentally measured values. Address potential quantitative discrepancies that may arise from the zero-temperature assumption in the simulations.
6. Boundary Conditions in Simulations: Justify the use of periodic boundary conditions in the micromagnetic simulations. Validate whether the implicit treatment of dipolar interactions accurately reflects the experimental multilayer structure.
7. Figure 3c: Replace the "Guide to eye" lines with theoretical fits or explicitly label them as trendlines to improve clarity and reduce subjectivity.
8. Supplementary Data: In Supplementary Figure 1, quantify the stability of AFM coupling before and after current pulsing (e.g., through magnetic contrast analysis) to substantiate claims of robustness.
9. References: Ensure that all relevant references, such as *National Science Review*, Vol. 6, No. 2, pp. 210-212 (2019), are cited appropriately within the manuscript to acknowledge prior work and provide context.

Addressing these points will significantly enhance the clarity, rigor, and impact of the manuscript.

Version 1:

Reviewer comments:

Reviewer #1

(Remarks to the Author)

Authors answered all my questions and questions from the other reviewer satisfactorily. In my opinion, this manuscript can now be published in Nature Communications.

Reviewer #2

(Remarks to the Author)

The authors have satisfactorily addressed most of the major comments raised by the reviewers. Therefore, I recommend that it be published in NC as is.

Response to Reviewer #1

We are grateful to the reviewer for the conscientious reading of our manuscript and the helpful suggestions. We are glad to hear the positive appreciation of our work, and we have revised the manuscript according to the comments and incorporated the suggestions. Our responses are listed below.

Reviewer's comment

In the work by Dohi et al., the authors have investigated the non-reciprocal transport phenomena of the 3D hybrid chiral skyrmion tubes in synthetic antiferromagnetic (SyAFM) multilayers. The authors attributed this non-reciprocal behavior to the asymmetric spin-orbit torque (SOT) effects that is associated with the dynamic changes of the skyrmion tube during its motion. Specifically, the helicity of the skyrmion evolves dynamically during motion, which results in the non-reciprocal behavior, rather than being caused by shape distortion. The experiments and simulations seem systematic. However, the current version of the manuscript is not suitable for publication in Nature Communications. I would encourage authors to revise the manuscript based on the following comments and extend necessary discussions and experiments. I would be happy to take a second look of the revised manuscript.

Our reply

We are pleased to learn that the reviewer recognizes the novelty of our manuscript as well as the well-organized experiments and simulations. Also, the reviewer is inclined to support its publication in Nature Communications in principle after a suitable revision. We highly appreciate the reviewer's valuable suggestions for further enhancing the quality of our work. In response, we have carried out significant additional work based on Lorentz-TEM, allowing us to directly visualize the subtle spin structure details as well as micromagnetic simulations, which all corroborate the explanation that we put forward in the manuscript.

Reviewer's comment

(1) On Page 4, the authors mentioned that the observed non-reciprocal skyrmion Hall effect is qualitatively different from the previously observed non-reciprocal dynamics of conventional 2D skyrmions with extrinsic in-plane symmetry breaking... However, no relevant references were inserted at this point, which makes this referee hard to follow.

Our reply

We thank the reviewer for this comment. We apologize that we previously provided detailed explanations with citations only in the later sections of the original version of the manuscript (313 lines, page 14). So, we agree that including references at the pointed-out location will improve readability. Following the reviewer's suggestion, we have added the relevant citations to the following passages.

[page 4, line 88-91]

<From>

and thus it is qualitatively different from the previously observed non-reciprocal dynamics of conventional 2D skyrmions with extrinsic in-plane symmetry breaking, e.g., from asymmetric confinements/pinning, or exchange bias

<To>

and thus it is qualitatively different from the previously observed non-reciprocal dynamics of conventional 2D skyrmions with extrinsic in-plane symmetry breaking, e.g., from asymmetric confinements/pinning³⁶, or exchange bias³⁷

Reviewer's comment

(2) On Page 5, what is the unit of thickness inside the parentheses in the stack structures of ST1 and ST2?

Our reply

We appreciate the reviewer's comment. Since the unit of film thickness is nm, we have added the following clarification.

[page 5, line 107-109]

<From>

The stack structures consist of Ta(5.00)/[Pt(1.00)/Ir(0.40)/FM_{b1}/Pt(1.00)/Ir(0.40)/FM_{t1}/Pt(1.00) (ST1, hereafter),

<To>

The stack structures consist of Ta(5.00)/[Pt(1.00)/Ir(0.40)/FM_{b1}/Pt(1.00)/Ir(0.40)/FM_{t1}]_{×15}/Pt(1.00) (numbers in nm) (ST1, hereafter),

Reviewer's comment

(3) On Page 6, the authors mentioned that the magnetic contrast in the STXM results represents the averaged response of all layers. How did the authors confirm the presence of skyrmions in each individual layer? How authors can rule out the existence of layer dependent skyrmionic textures, which is typical in skyrmion hosting multilayers?

Our reply

We thank the reviewer for this insightful comment. STXM provides a depth-averaged projection of the magnetic contrast and, on its own, cannot unambiguously distinguish between extended skyrmion tubes and more complex three-dimensional spin textures such as chiral bobbbers, cocoons, or Bloch-point terminations. However, the absolute grey scale level is a very clear indicator of whether the skyrmions are present in all layers. So, it is important to note that three-dimensional textures, particularly those that do not span the full multilayer thickness, will produce STXM contrast with multiple gray levels, rather than the bimodal contrast associated with uniform skyrmion tubes. This occurs because the magnetic signal is averaged through the thickness, and partial or non-uniform structures (such as chiral cocoons or bobber-style terminated skyrmions) contribute differently to the total magnetization along the beam direction. Such multi-level contrast has been used as a signature of vertical inhomogeneity. These 3D textures typically arise in specially engineered heterostructures where magnetic anisotropy or DMI varies significantly across the film thickness, or where dipolar interactions dominate the magnetic energy landscape. These types of 3D textures generally emerge in specially engineered heterostructures, where magnetic anisotropy or Dzyaloshinskii–Moriya interaction varies significantly with depth, or in systems where dipolar interactions dominate due to weak interlayer coupling. In contrast, the multilayer stacks investigated in our study were specifically designed to exhibit strong interlayer antiferromagnetic exchange coupling, mediated by ultrathin Ir spacers (see Figure 1a). Furthermore, the anisotropy of alternating magnetic layers was carefully engineered to ensure a uniform effective anisotropy across the stack. This strong interlayer coupling dominates over local anisotropy fluctuations and energetically favors coherent spin textures that persist uniformly throughout the multilayer. So as a first indication, we see that in our case, we see only single white-black contrast amplitudes, showing that there are no different grey levels.

To further directly test the effect of interlayer exchange, we prepared a comparison sample with significantly weaker coupling. When we deliberately increased the thickness of a Pt insertion layer to weaken the interlayer coupling (by a factor of ~ 4), we observed pronounced heterogeneity in the magnetic contrast (see Figure R1), consistent with the emergence of three-dimensional spin textures. The schematic of the stack and its magnetic hysteresis loop is shown in Figure R2. In contrast, the samples investigated in this study, as evidenced by Figure 1 and the Supplementary Movie, retain uniformly strong interlayer exchange and exhibit homogeneous skyrmion contrast, with no indication

of heterogeneous contrast from three-dimensional textures. These observations demonstrate that the layer-specific magnetic configurations suggested by the reviewer do not occur in our films. We now mention this in the revised manuscript as below.

Figure R1 Formation of three-dimensional spin textures in weakly-exchange-coupled SyAFM systems leading to different greyscale levels.

Figure R2 (a) Schematic illustration of the multilayer stack with reduced interlayer exchange coupling, achieved by inserting a thicker Pt spacer layer. (b) Magnetic hysteresis loop of the corresponding stack, measured using SQUID magnetometry, confirming the weakened interlayer coupling suggested by the lower saturation fields.

[page 6, line 139-142]

<Added>

Furthermore, all skyrmion tubes exhibit no discernible multilevel grey XMCD contrast, confirming that inhomogeneous skyrmionic textures such as skyrmionic cocoons or bobber-style terminated skyrmions are absent owing to strong interlayer exchange coupling.

Reviewer's comment

(4) The authors should include some images of skyrmion motion in the main text to help readers better visualize and understand the experimental phenomena.

Our reply

We thank the reviewer for this valuable suggestion. To enhance the clarity of our experimental methods and results, we have revised the corresponding Figure as shown below and updated its corresponding Fig. numbers.

[Figure 2]

<From>

Figure 2

Current-induced motion of SyAFM skyrmion tubes. (a) The full statistics of the individual SyAFM skyrmion tube velocity for the stack ST1 ($m_c = 50\%$) under $\mu_0 H_z = 70 \text{ mT}$ at $J = 13 \times 10^{11} \text{ A m}^{-2}$. Red and blue colors correspond to the positive and negative current polarity, respectively. The curves denote the Gaussian curve fitting. (b) The current density dependence of the average velocity, v_{ave} , is defined by the Gaussian peak with $\pm 1\sigma$ bars of the curve fitting in Fig.2a. Red circles, blue rectangles, magenta triangles, and orange inverse triangles represent positive current polarity (ST1 stack), negative polarity (ST1 stack), positive polarity (ST2 stack), and negative polarity (ST2 stack), respectively.

Figure 2

Current-induced motion of SyAFM skyrmion tubes. (a) Representative tracked trajectories of current-induced motion of SyAFM skyrmion tubes. (b) The full statistics of the individual SyAFM skyrmion tube velocity for the stack ST1 ($m_C = 50\%$) under $\mu_0 H_z = 70$ mT at $J = 13 \times 10^{11} \text{ A m}^{-2}$. Red and blue colors correspond to the positive and negative current polarity, respectively. The curves denote the Gaussian curve fitting. (c) The current density dependence of the average velocity, v_{ave} , is defined by the Gaussian peak with $\pm 1\sigma$ bars of the curve fitting in Fig.2b. Red circles, blue rectangles, magenta triangles, and orange inverse triangles represent positive current polarity (ST1 stack), negative polarity (ST1 stack), positive polarity (ST2 stack), and negative polarity (ST2 stack), respectively.

Reviewer's comment

(5) In Figure 3b, when the average skyrmion velocity reaches 10 m/s, the skyrmion Hall angles under positive and negative currents show a significant difference, which does not consistent with Figure 3a. The authors should provide an explanation.

Our reply

We thank the referee for this comment. In fact, Figs 3a and 3b are fully consistent but apparently we did not explain this well enough. In Fig. 3a, the precise velocities used for evaluating the skyrmion Hall angle are 11.0 m/s (positive current pulse) and 11.2 m/s (negative current pulse). At these velocities, the peaks of the Gaussian fits in Figure 3a occur at 43.6° and 38.7° , respectively. In this statistical dataset (Figure 3a), each bin on the horizontal axis spans 4° . Consequently, the observed 4.9° difference corresponds to approximately one bin width, rendering it imperceptible by eye and indicating no significant θ_{ave} difference between positive and negative current polarities. Moreover, these values completely match the data shown in Figure 3b around 10 m/s. This is now better explained in the revised manuscript.

[METHODS section: **Element-specific magnetic domain observation and current-induced dynamics**, page 29, line 623-624]

<Added>

In the statistical dataset corresponding to Figure 3a, each bin on the horizontal axis spans 4° , reflecting the resolution of this dataset.

Reviewer's comment

(6) The authors should provide the definitions of “creep/depinning regime” and “flow regime” in the main text. Please also clarify how authors identify these regimes.

Our reply

We thank the referee for this comment. We define the region in which the skyrmion velocity scales linearly with current as the flow regime, and the region exhibiting nonlinear behavior as the creep/depinning regime. To make these definitions explicit, we have revised Fig. 2c in the updated manuscript by adding gray solid trendlines that indicate the expected linear response at each current. From this modified Fig. 2, we determine that the $m_C \approx 80\%$ sample has a threshold velocity of approximately 25 m/s, whereas the $m_C \approx 50\%$ sample shows a threshold near 10 m/s.

[Figure 2]

<From>

Figure 2

Current-induced motion of SyAFM skyrmion tubes. (a) The full statistics of the individual SyAFM skyrmion tube velocity for the stack ST1 ($m_C = 50\%$) under $\mu_0 H_z = 70$ mT at $J = 13 \times 10^{11}$ Am $^{-2}$. Red and blue colors correspond to the positive and negative current polarity, respectively. The curves denote the Gaussian curve fitting. (b) The current density dependence of the average velocity, v_{ave} , is defined by the Gaussian peak with $\pm 1\sigma$ bars of the curve fitting in Fig.2a. Red circles, blue rectangles, magenta triangles, and orange inverse triangles represent positive current polarity (ST1 stack), negative polarity (ST1 stack), positive polarity (ST2 stack), and negative polarity (ST2 stack), respectively.

<To>

Figure 2

Current-induced motion of SyAFM skyrmion tubes. (a) Representative trajectory tracking of current-induced motion of SyAFM skyrmion tubes. (b) The full statistics of the individual SyAFM skyrmion tube velocity for the stack ST1 ($m_C = 50\%$) under $\mu_0 H_z = 70$ mT at $J = 13 \times 10^{11}$ A m⁻². Red and blue colors correspond to the positive and negative current polarity, respectively. The curves denote the Gaussian curve fitting. (c) The current density dependence of the average velocity, v_{ave} , is defined by the Gaussian peak with $\pm 1\sigma$ bars of the curve fitting in Fig.2b. Red circles, blue rectangles, magenta triangles, and orange inverse triangles represent positive current polarity (ST1 stack), negative polarity (ST1 stack), positive polarity (ST2 stack), and negative polarity (ST2 stack), respectively. **In line with literature, we use the common definition of the flow regime as the region in which the average skyrmion velocity exhibits a linear dependence on the applied current density.**

Reviewer's comment

(7) The authors proposed that the non-reciprocal skyrmion Hall effect is closely related to the degree of compensation. Could the authors provide a micromagnetic simulation result showing how the skyrmion Hall effect changes from a partly compensated (FIM) state, through various degrees of compensation, to a fully compensated (AFM) state?

Our reply

We thank the reviewer for this insightful suggestion. The related dynamics of hybrid skyrmions and their associated non-reciprocal Hall effect have been discussed previously using micromagnetic simulations in the literature e.g., Phys. Rev. B **97**, 224427 (2018), where non-reciprocal motion was observed across a broad range of current densities. In our SyAFM system, the degree of magnetic compensation plays a central role in stabilizing hybrid skyrmions. At partial compensation (e.g., ~50%), skyrmions are stabilized via a competition between Dzyaloshinskii-Moriya interaction and long-range dipolar interactions. This results in hybrid skyrmions with non-uniform spin textures along the thickness, as supported by recent Lorentz TEM imaging (additionally realized measurements for this review process) and micromagnetic simulations.

To address the reviewer's comment, we have extended our micromagnetic simulations to include compensation levels of 20%, 40%, 60%, and 80%, in addition to the previously studied cases, which we added as Supplementary Note 6 in the revised manuscript. At 20% compensation, where the system forms a hybrid skyrmion, we observe a pronounced non-reciprocal skyrmion Hall effect as expected. As compensation increases and the net moment is suppressed, this non-reciprocity gradually diminishes. At 80% compensation, the skyrmion profile becomes Néel-type, and the Hall angle asymmetry disappears, consistent with the expectation for symmetric spin structures and reduced net

moment. We have added the additional Supplementary Note 6 that provides a clearer physical picture to elucidate the underlying physics for our experimental observations.

Supplementary Note 6

Systematic micromagnetic simulation:

magnetic compensation dependence of the static and dynamic properties

Here, we investigate how the magnetic compensation affects both the helicity distribution through the film thickness and the current-driven skyrmion dynamics, thereby elucidating the underlying physics. Supplementary Figure 6a plots the equilibrium helicity in the topmost and bottommost layers of antiferromagnetically coupled skyrmion tubes as a function of compensation. At 80 % and 60 % compensation, the helicity is uniform across all layers, corresponding to a pure Néel-type configuration. At 50 % compensation, a helicity gradient arises, yielding a hybrid Néel-Bloch character. As compensation falls to 40 %, the Bloch component grows more pronounced, and at 20 %, the domain-wall texture approaches a nearly pure Bloch-type, as expected.

Supplementary Figures 6b-e show the corresponding current-induced dynamics. At 20 % compensation, a pronounced non-reciprocal skyrmion Hall effect appears due to the large helicity difference between the top and bottom layers. As compensation increases, this non-reciprocal response is observed only at higher current densities. At 60 % and 80 % compensation, the skyrmion Hall angle is symmetric with respect to current polarity, consistent with a pure Néel-type structure lacking Bloch components, and a non-reciprocal effect becomes insignificant.

Supplementary Figure 6 | Systematic micromagnetic simulations of antiferromagnetic hybrid chiral skyrmion tubes. **a**, 2D magnetization profiles of the topmost and bottommost layers for skyrmions in SyAFM stacks with 80 %, 60 %, 50 %, 40 %, and 20 % magnetic compensation, respectively. Arrow directions indicate the magnetization orientation within each layer, while color variations denote the magnetization direction along the x -axis. The skyrmion Hall angle as a function of current density for varying degrees of magnetic compensation: **b**, 20%, **c**, 40%, **d**, 60%, and **e**, 80%.

[METHODS section: **Micromagnetic simulations**, page 30, line 648-650]

<To>

To clarify the underlying physical picture, we performed a systematic and detailed investigation of the magnetic compensation dependence, as presented in Supplementary Note 6.

Reviewer's comment

(8) The authors mention that the non-reciprocal skyrmion Hall effect is also closely related to the "hybridness" of skyrmion tubes. If authors wanted to bridge this hybridness with hybrid skyrmion in multilayers, I would like to see the evidence of hybrid skyrmions. In addition, authors should explain how to precisely control the "hybridness" of skyrmion tubes experimentally?

Our reply

We thank the referee for this valuable suggestion. While achieving true layer-by-layer imaging of our synthetic antiferromagnetic stacks remains extremely challenging, we have, in response, now carried out Lorentz transmission electron microscopy (Lorentz TEM) on the 50 %-compensated multilayer sample to reveal the subtle Bloch component indicative of hybrid character. Because Lorentz TEM is a transmission-based technique, the resulting contrast represents the average in-plane magnetization through the film thickness. In these measurements, we observe signatures of both Néel and Bloch components, which supports the presence of hybrid skyrmions. However, we note that Lorentz TEM will not resolve individual layers, so it is not possible to determine whether the Bloch component is uniformly distributed across the stack or localized in specific layers. However, given the contrast analysis, we have evidence that the spin structures are largely uniform across the thickness. We have now made this limitation explicit in the revised Supplementary Note 2 and present the relevant data in Supplementary Fig. 2.

Regarding the control of hybrid character, we emphasize that it is governed primarily by the degree of magnetic compensation between layers, as now substantiated by the additional Supplementary Note 6. By adjusting the relative thickness and magnetic moments of the ferromagnetic sublayers, we systematically vary the net magnetization. This variation alters the balance between interfacial Dzyaloshinskii-Moriya interactions and long-range dipolar coupling, and thus modulates the hybrid nature of the skyrmions. The new supplementary sections now read

Supplementary Note 2

Lorentz transmission electron microscopy imaging

Here, we provide direct evidence of the presence of the Bloch components of domain walls in the

ST1 stack to substantiate our discussion and explanations in the main text. The Lorentz transmission electron microscopy (TEM) imaging was carried out on a 50 % compensated SAF deposited onto a SiN membrane using a TFS Titan transmission electron microscope operated at 300 kV in Lorentz mode. The objective lens was used to apply a magnetic field along the electron beam direction that was pre-calibrated using a Hall sensor. Images were acquired using a Gatan UltraScan 1000XP CCD camera.

Supplementary Figure 2a and b are Lorentz TEM images obtained at a tilt angle of 0° and 10° , respectively, that show stripe domains. In out-of-plane magnetized samples, the contrast at 0° depends on the type of domain wall. Bloch domain walls produce contrast, and the Néel-type do not. In a tilted condition, the contrast depends on both the domain wall type and the projection of the out-of-plane component of the magnetic field in the image plane⁴. Here, the magnetic contrast is particularly weak at 0° compared to 10° , which indicates that the magnetic texture is essentially Néel-type. To obtain more information, profiles were extracted from the images across domains as shown in supplementary Fig. 2c and d. In supplementary Fig. 2c, the profiles show a bright/dark/bright distribution as indicated by three arrows in the plots. This distribution indicates the presence of a small Bloch component⁵. In supplementary Fig. 2d, the blue profile was extracted from a domain oriented perpendicular to the tilt axis, which shows a strong dark/bright distribution, as expected from the projection of the out-of-plane component. The red profile in supplementary Fig. 2d was extracted from a domain oriented parallel to the tilt axis. It shows again a weak bright/dark/bright distribution similar to that obtained at 0° . This is another indication of the presence of a small Bloch component. At a tilt angle of 10° , the projection of the out-of-plane magnetization along the imaging direction is approximately $\sin 10^\circ \approx 17\%$. Given that the Bloch component contributes to only one-sixth of the full contrast under such geometry, its relative contribution is estimated to be $\sim 3\%$.

Supplementary Figure 2 | Lorentz TEM images obtained with a defocus of 4 mm, an external magnetic field of 85 mT, and a tilt angle of **a**, $\alpha = 0^\circ$, and **b**, $\alpha = 10^\circ$. The non-magnetic background was subtracted using images acquired after saturation⁶. Upon tilting, the domain contrast is enhanced for those aligned along the tilt axis, indicating the presence of in-plane magnetization components. **c**, and **d**, line profiles extracted from the blue and red lines in panels **a** and **b**, respectively. The polarity and asymmetry in the contrast across individual domain walls show the mixed Néel-Bloch character, consistent with hybrid skyrmions that lend themselves to the observed effects.

We have also added additional sentences below in the main text to cite this new Supplementary Note 2, as well as updated the author lists to include collaborators who helped us to characterize the hybrid character by Lorentz TEM imaging.

[page 10, line 221-224]

<From>

In contrast, at 50 % compensation, a Néel-type domain wall characterizes one end of the skyrmion tube with helicity (η) being zero, while the helicity varies continuously through successive layers, maintaining AFM alignment at their cores.

<To>

In contrast, at 50 % compensation, a Néel-type domain wall characterizes one end of the skyrmion tube with helicity (η) being zero, while the helicity varies continuously through successive layers, maintaining AFM alignment at their cores. The simulation also indicates that the hybrid character should have a subtle but finite Bloch component, which is evidenced by Lorentz transmission microscope imaging (details, see Supplementary Note 2).

Reviewer's comment

(9) In the magnetic multilayers with low values of DMI, due to the influence of dipolar interactions, different chiral spin structures may appear in the top and bottom layers. Could the authors add the simulation results for the top layer (layer 15 for ST1 and layer 25 for ST2) in Figure 4a?

Our reply

We thank the referee for this comment, and we apologize for any confusion caused by our Fig. 4. Figure 4 already represents the topmost layer of the multilayer stack. To keep computational costs manageable, we simulated the dynamics in the maximum number of layers that still allowed convergence. We note, however, that this choice may underestimate dipole-dipole interactions, and therefore our discussion of the non-reciprocal skyrmion Hall effect (NSkHE) remains qualitative. As shown in Supplementary Note 6, our additional simulations reveal that the NSkHE becomes more pronounced when the skyrmion hybrid character is stronger, which corresponds to the dominance of dipolar interactions. Direct comparison of the simulation results with the experimental data in Figs 2 and 3 of the main text confirms that our qualitative description of the NSkHE captures the observed behavior.

We have also made explicit in the newly added Supplementary Note 6 how the chiral spin structure at the topmost and bottommost layers varies significantly with magnetic compensation and how this variation correlates with the NSkHE, as the reviewer suggested. We believe that these additions provide a clearer and more complete physical picture.

Supplementary Note 6

Systematic micromagnetic simulation:

magnetic compensation dependence of the static and dynamic properties

Here, we investigate how the magnetic compensation affects both the helicity distribution through the film thickness and the current-driven skyrmion dynamics, thereby elucidating the underlying physics. Supplementary Figure 6a plots the equilibrium helicity in the topmost and bottommost layers

of antiferromagnetically coupled skyrmion tubes as a function of compensation. At 80 % and 60 % compensation, the helicity is uniform across all layers, corresponding to a pure Néel-type configuration. At 50 % compensation, a helicity gradient arises, yielding a hybrid Néel-Bloch character. As compensation falls to 40 %, the Bloch component grows more pronounced, and at 20 %, the domain-wall texture approaches a nearly pure Bloch-type, as expected.

Supplementary Figures 6b-e show the corresponding current-induced dynamics. At 20 % compensation, a pronounced non-reciprocal skyrmion Hall effect appears due to the large helicity difference between the top and bottom layers. As compensation increases, this non-reciprocal response is observed only at higher current densities. At 60 % and 80 % compensation, the skyrmion Hall angle is symmetric with respect to current polarity, consistent with a pure Néel-type structure lacking Bloch components, and a non-reciprocal effect becomes insignificant.

Supplementary Figure 6 | Systematic micromagnetic simulations of antiferromagnetic hybrid chiral skyrmion tubes. **a**, 2D magnetization profiles of the topmost and bottommost layers for skyrmions in SyAFM stacks with 80 %, 60 %, 50 %, 40 %, and 20 % magnetic compensation, respectively. Arrow directions indicate the magnetization orientation within each layer, while color variations denote the magnetization direction along the x -axis. The skyrmion Hall angle as a function of current density for varying degrees of magnetic compensation: **b**, 20%, **c**, 40%, **d**, 60%, and **e**, 80%.

We have also explicitly stated this limitation in the **METHODS** section in the main text.

[**METHODS** section: **Micromagnetic simulations**, page 30, line 646-648]

<Added>

Note that in order to keep computational cost manageable, we used twelve repetitions of the multilayer stack, which was the maximum number allowing for reasonable convergence.

Reviewer's comment

(10) The labels in Figure 4 are incorrect. The authors should carefully check the manuscript to avoid discrepancies between the main text and the figures.

Our reply

We thank the reviewer for bringing this to our attention. We have carefully reviewed the manuscript several times and corrected all typographical errors, as documented in the tracked changes. However, we have been unable to locate the mislabeled element in Figure 4. Would the reviewer kindly specify the panel or annotation in question? We apologize for this oversight and appreciate any guidance to help us rectify the issue.

Response to Reviewer #2

We are grateful to the reviewer for the critical and conscientious reading of our manuscript. In line with reviewer #1, the positive evaluation of our work that warrants publication in Nature Communications is highly appreciated. We have revised the manuscript according to the comments and suggestions. In particular in response to the comments, we have carried out significant additional experimental work using Lorentz-TEM and carried out additional simulations and all the results further corroborate our interpretation. Our detailed responses are listed below.

Reviewer's comment

This manuscript presents the observation of a non-reciprocal skyrmion Hall effect (NSkHE) in hybrid chiral skyrmion tubes within synthetic antiferromagnetic (SyAFM) multilayers. By combining experimental characterization with micromagnetic simulations, the study highlights the critical role of dynamic helicity oscillations in this phenomenon. The work is of significant scientific merit, offering valuable insights into the dynamics of 3D topological spin textures and their potential applications in spintronics. However, the following issues need to be addressed or clarified before the manuscript can be considered for acceptance:

Our reply

We are pleased to learn that the reviewer has recognized the novelty and importance of our work and, in principle, supported the publication of our manuscript in Nature Communications. We also thank the reviewer for summarizing the key points of our work.

Reviewer's comment

1. Figures 2a and 3a: Include goodness-of-fit metrics (e.g., R^2 values) for the Gaussian fits. Additionally, provide data from repeated experiments to confirm the reproducibility of the results.

Our reply

We appreciate the reviewer's suggestion. We have now included the R^2 values for all statistical analyses; in each case, the fits yield $R^2 > 0.8$, thereby demonstrating the reliability of our results. Accompanying these additions, we have revised Figures 2 and 3 to display the updated R^2 annotations.

Regarding reproducibility, we performed measurements on a number of devices of varying sizes and incorporated all datasets into our statistical analysis. Consequently, the observed trends are not based on a single device but are consistently reproduced across multiple devices, confirming the

robustness of our findings.

[Figure 2]

<From>

<To>

[Figure 3]

<From>

<To>

Reviewer's comment

2. Skyrmion Diameter Stabilization: Clarify the mechanism and range of stabilization for skyrmion diameters (~100 nm) under out-of-plane fields. Discuss how variations in skyrmion size ($\pm 15\%$) may influence the NSkHE and its dynamics.

Our reply

We thank the referee for the valuable comment. In multilayer thin-film systems, the stabilization of skyrmions is governed by the delicate interplay of competing magnetic interactions, including interfacial Dzyaloshinskii-Moriya interaction, interlayer exchange, perpendicular magnetic anisotropy, exchange stiffness, and long-range dipolar interactions. In such a system, the energetic conditions for skyrmion formation can be broadly categorized into two regimes:

1. **Metastable regime:** When the domain wall (DW) energy is positive, the uniform magnetic state is the ground state. Nevertheless, skyrmions may still be stabilized as metastable excitations under suitable external stimuli, such as magnetic fields or spin-orbit torques, even in the absence of an external bias.
2. **Spontaneous regime near spin reorientation transition (SRT):** In systems where the effective anisotropy approaches zero ($K_{\text{eff}} \approx 0$), the DW energy, $\varepsilon_{\text{Néel}} = 4\sqrt{A_S K_{\text{eff}}} - \pi D_i$ can become negative. This indicates an instability of the uniform state and promotes the spontaneous emergence of chiral spin textures. In this regime, iDMI and dipolar interactions dominate over the stabilizing exchange and anisotropy energies, driving the system into a multidomain ground state that can include skyrmions.

In our SyAFM multilayers, we utilize this latter regime by tuning each FM layer thickness to bring the system close to the SRT, thus reducing the effective anisotropy. This results in a spontaneously formed multidomain state, from which skyrmions can be selectively stabilized by applying an out-of-plane magnetic field. The field energetically penalizes antiparallel domains and enables skyrmion stabilization within a finite field window. Due to the near-degenerate energy landscape, where circular skyrmions and stripe domains coexist, external stimuli are often required to overcome the topological barrier associated with skyrmion nucleation. We employ two approaches to initialize skyrmion states: (1) out-of-plane magnetic field cycling, (2) current-induced nucleation. These methods allow us to prepare well-defined initial states for probing the dynamical behavior of skyrmions across different compensation levels.

Long-range dipolar interactions play a critical role in multilayer thin films, where the system tends to minimize magnetostatic energy by forming flux-closure configurations, facilitating the emergence of non-uniform textures such as hybrid skyrmions. In synthetic antiferromagnets (SyAFMs), the strength of these dipolar interactions can be tuned by varying the degree of magnetic compensation. In our 50 % compensated SyAFM samples, we observe the formation of hybrid skyrmions. These hybrid skyrmions show a distinct dynamic behavior in comparison to the Néel skyrmions, as evidenced by our Lorentz TEM imaging additionally implemented in this review process. In our system, skyrmions typically exhibit diameters of approximately 100 nm, with an experimental uncertainty of ± 20 nm (within the range of 15-20% size variation) arising from the spatial resolution of STXM. Unlike metastable skyrmions, their size remains largely invariant under increased magnetic field, as the stripe domain periodicity shows a similar length scale as skyrmions in our systems. Instead of shrinking/expanding, the skyrmions undergo a transition into a monodomain/stripe-like state. Also, we note that as the non-reciprocal skyrmion Hall effect depends on helicity modulation rather than skyrmion shape distortion, the slight variations in skyrmion size do not produce measurable changes in the effect. We have modified this in the method section of the manuscript.

[METHODS section: **Element-specific magnetic domain observation and current-induced dynamics**, page 28-29, line 601-614]

<Added>

Skyrmion tubes in our system nucleate spontaneously near the spin reorientation transition (SRT), where the effective anisotropy approaches zero and may become negative. Under these conditions, the uniform magnetic state becomes unstable, and spin textures including chiral skyrmions are favored. In this regime, the interfacial Dzyaloshinskii-Moriya interaction and long-range dipolar interactions outweigh the exchange and anisotropy energies, driving the formation of a multidomain ground state that can include stripe domains and skyrmions. By adjusting the thickness of each ferromagnetic layer in our synthetic antiferromagnetic multilayers, we tune the system into this SRT regime and thus promote spontaneous multidomain formation. Skyrmions are selectively stabilized by applying an out-of-plane magnetic field, which penalizes antiparallel domain configurations and favors isolated circular textures. As the energy landscape is nearly degenerate, with circular skyrmions and stripe domains coexisting, external stimuli are required to overcome the nucleation barrier. To initialize well-defined skyrmion states across different compensation levels, we employ two methods: (1) out-of-plane magnetic field cycling and (2) current-induced nucleation.

Reviewer's comment

3. Experimental Evidence of Helicity Oscillations: While micromagnetic simulations suggest helicity oscillations, direct experimental evidence (e.g., time-resolved, layer-specific magnetic imaging via STXM or Lorentz microscopy) is absent. If feasible, include such data; otherwise, elaborate on how the simulations align with experimental observations to strengthen the interpretation.

Our reply

We thank the referee for this comment and suggestion. While we agree that time-resolved, layer-specific magnetic imaging (e.g., via STXM or Lorentz TEM) would provide direct evidence of helicity oscillations, such measurements are currently not feasible in our system. The skyrmion dynamics exhibit a strong non-reciprocal skyrmion Hall effect (SkHE), resulting in trajectories that do not return to the same position even after a pair of opposite current pulses. This lack of reproducibility makes pump-probe imaging approaches impossible, as they rely on the accumulation of repeatable motion over many cycles. Nevertheless, the helicity oscillations observed in our micromagnetic simulations are consistent with experimentally measured velocity asymmetries and the non-reciprocal nature of the skyrmion Hall effect at partial compensation.

However, to demonstrate that in our system the spin texture details that lend themselves to the non-reciprocal dynamics are present, we performed Lorentz TEM imaging. This allows us to obtain direct evidence of the hybrid character of the skyrmion tube that is key to driving the non-reciprocal skyrmion Hall effect. In the sample with 50 % magnetic compensation, we have captured the subtle signals arising from Bloch components that provide clear evidence of hybrid character. These results have been added as Supplementary Note 6 below. We believe that this addition further substantiates our conclusions.

Supplementary Note 2

Lorentz transmission electron microscopy imaging

Here, we provide direct evidence of the presence of the Bloch components of domain walls in the STI stack to substantiate our discussion and explanations in the main text. The Lorentz transmission electron microscopy (TEM) imaging was carried out on a 50 % compensated SAF deposited onto a SiN membrane using a TFS Titan transmission electron microscope operated at 300 kV in Lorentz mode. The objective lens was used to apply a magnetic field along the electron beam direction that was pre-calibrated using a Hall sensor. Images were acquired using a Gatan UltraScan 1000XP CCD camera.

Supplementary Figure 2a and b are Lorentz TEM images obtained at a tilt angle of 0° and 10°, respectively, that show stripe domains. In out-of-plane magnetized samples, the contrast at 0° depends on the type of domain wall. Bloch domain walls produce contrast, and the Néel-type do not. In a tilted

condition, the contrast depends on both the domain wall type and the projection of the out-of-plane component of the magnetic field in the image plane⁴. Here, the magnetic contrast is particularly weak at 0° compared to 10° , which indicates that the magnetic texture is essentially Néel-type. To obtain more information, profiles were extracted from the images across domains as shown in supplementary Fig. 2c and d. In supplementary Fig. 2c, the profiles show a bright/dark/bright distribution as indicated by three arrows in the plots. This distribution indicates the presence of a small Bloch component⁵. In supplementary Fig. 2d, the blue profile was extracted from a domain oriented perpendicular to the tilt axis, which shows a strong dark/bright distribution, as expected from the projection of the out-of-plane component. The red profile in supplementary Fig. 2d was extracted from a domain oriented parallel to the tilt axis. It shows again a weak bright/dark/bright distribution similar to that obtained at 0° . This is another indication of the presence of a small Bloch component. At a tilt angle of 10° , the projection of the out-of-plane magnetization along the imaging direction is approximately $\sin 10^\circ \approx 17\%$. Given that the Bloch component contributes to only one-sixth of the full contrast under such geometry, its relative contribution is estimated to be $\sim 3\%$.

Supplementary Figure 2 | Lorentz TEM images obtained with a defocus of 4 mm, an external magnetic field of 85 mT, and a tilt angle of **a**, $\alpha = 0^\circ$, and **b**, $\alpha = 10^\circ$. The non-magnetic background was subtracted using images acquired after saturation⁶. Upon tilting, the domain contrast is enhanced

for those aligned along the tilt axis, indicating the presence of in-plane magnetization components. **c**, and **d**, line profiles extracted from the blue and red lines in panels **a** and **b**, respectively. The polarity and asymmetry in the contrast across individual domain walls show the mixed Néel-Bloch character, consistent with hybrid skyrmions that lend themselves to the observed effects.

We have also added relevant sentences below in the main text to cite this new Supplementary Note 2, as well as updated the author lists to include collaborators who helped us to characterize the hybrid character by Lorentz TEM imaging.

[page 10, line 221-224]

<From>

In contrast, at 50 % compensation, a Néel-type domain wall characterizes one end of the skyrmion tube with helicity (η) being zero, while the helicity varies continuously through successive layers, maintaining AFM alignment at their cores.

<To>

In contrast, at 50 % compensation, a Néel-type domain wall characterizes one end of the skyrmion tube with helicity (η) being zero, while the helicity varies continuously through successive layers, maintaining AFM alignment at their cores. The simulation also indicates that the hybrid character should have a subtle but finite Bloch component, which is evidenced by Lorentz transmission microscope imaging (details, see Supplementary Note 2).

Reviewer's comment

4. Magnetic Compensation: The NSkHE is observed exclusively at 50% magnetic compensation. Discuss whether this phenomenon could extend to other compensation levels or material systems, such as ferromagnetic multilayers.

Our reply

We thank the referee for this insightful comment. While our work focuses on partially compensated SyAFM multilayers, the underlying physics arising from the interplay between DMI and dipolar interactions can extend to other compensation levels and even to ferromagnetic multilayers. To assess the generality of our findings, we performed additional micromagnetic simulations at 20 %, 40 %, 60 %, and 80 % magnetic compensation. These simulations confirm the presence of a non-reciprocal skyrmion Hall effect across all applied current densities for lower magnetic compensation less than 20 %, consistent with the mechanism proposed in the main text. We believe that our additional

systematic simulation results aid readers in understanding the underlying physics of the non-reciprocal skyrmion Hall effect. The results are provided additionally as a Supplementary Note 6 presented below.

Supplementary Note 6

Systematic micromagnetic simulation:

magnetic compensation dependence of the static and dynamic properties

Here, we investigate how the magnetic compensation affects both the helicity distribution through the film thickness and the current-driven skyrmion dynamics, thereby elucidating the underlying physics. Supplementary Figure 6a plots the equilibrium helicity in the topmost and bottommost layers of antiferromagnetically coupled skyrmion tubes as a function of compensation. At 80 % and 60 % compensation, the helicity is uniform across all layers, corresponding to a pure Néel-type configuration. At 50 % compensation, a helicity gradient arises, yielding a hybrid Néel-Bloch character. As compensation falls to 40 %, the Bloch component grows more pronounced, and at 20 %, the domain-wall texture approaches a nearly pure Bloch-type, as expected.

Supplementary Figures 6b-e show the corresponding current-induced dynamics. At 20 % compensation, a pronounced non-reciprocal skyrmion Hall effect appears due to the large helicity difference between the top and bottom layers. As compensation increases, this non-reciprocal response is observed only at higher current densities. At 60 % and 80 % compensation, the skyrmion Hall angle is symmetric with respect to current polarity, consistent with a pure Néel-type structure lacking Bloch components, and a non-reciprocal effect becomes insignificant.

Supplementary Figure 6 | Systematic micromagnetic simulations of antiferromagnetic hybrid chiral skyrmion tubes. a, 2D magnetization profiles of the topmost and bottommost layers for skyrmions in SyAFM stacks with 80 %, 60 %, 50 %, 40 %, and 20 % magnetic compensation, respectively. Arrow directions indicate the magnetization orientation within each layer, while color variations denote the magnetization direction along the x -axis. The skyrmion Hall angle as a function of current density for varying degrees of magnetic compensation: **b,** 20%, **c,** 40%, **d,** 60%, and **e,** 80%.

Reviewer's comment

5. Simulation Parameters: Explicitly state whether the damping constant ($\alpha = 0.1$) and interfacial DMI strength ($D = 0.45 \text{ mJ/m}^2$) used in the simulations correspond to experimentally measured values. Address potential quantitative discrepancies that may arise from the zero-temperature assumption in the simulations.

Our reply

We thank the reviewer for the insightful comment regarding the simulation parameters. We reasonably estimated the interfacial Dzyaloshinskii-Moriya interaction strength based on our prior experimental observations [M. Bhukta *et al.*, Nat. Commun. **15**, 1641 (2024)]. By contrast, the magnetic damping parameter was determined directly in this study. We have updated the main text to clarify this distinction.

The simulations were performed under a zero-temperature assumption with experimentally obtained magnetic parameters, which inherently neglects thermal fluctuations present at room temperature in experiments. Additionally, our experimental multilayer stacks exhibit variations in magnetic and structural properties, as well as thermal fluctuations, as the reviewer noted. These factors give rise to the observed Gaussian distribution and statistical uncertainty in the skyrmion Hall angle, thereby explaining some of the quantitative discrepancies between experiment and simulation. We therefore emphasize, as stated in lines 294 – 297 of the manuscript, that the simulation results are intended to provide qualitative insights into the skyrmion dynamic behaviors, rather than quantitatively reproducing experimental measurements.

[METHODS section: **Magnetic parameters**, page 28, line 582-588]

<From>

The magnitude of the exchange stiffness A_S and the interfacial DMI D_i were taken to be 10 pJm^{-1} , and $0.4\text{-}0.5 \text{ mJm}^{-1}$ based on our previous observations¹⁸, which reproduces reasonably well the experimentally observed size of the SyAFM skyrmion tubes. A microwave-excited FMR and ST-FMR were used for quantitatively assessing the interlayer exchange coupling fields $\mu_0 H_{\text{ex}} \sim 0.2 \text{ T}$, the effective damping constant $\alpha = 0.1$, and the effective spin Hall angle $\theta_{\text{SH}} = 0.1$ in the SyAFM system^{55,56} (see Supplementary Note 3 for details).

<To>

The magnitude of the exchange stiffness A_S and the interfacial DMI D_i were **estimated** to be 10 pJ m^{-1} , and $0.4\text{-}0.5 \text{ mJ m}^{-1}$ based on our previous observations¹⁸, which reproduces reasonably well the experimentally observed size of the SyAFM skyrmion tubes. Microwave-excited FMR and ST-FMR measurements were used for **experimentally obtaining** the interlayer exchange coupling fields $\mu_0 H_{\text{ex}} \sim 0.2 \text{ T}$, the effective damping constant $\alpha = 0.1$, and the effective spin Hall angle $\theta_{\text{SH}} = 0.1$ in the SyAFM

system^{55,56} (see Supplementary Note 3 for details).

Reviewer's comment

6. Boundary Conditions in Simulations: Justify the use of periodic boundary conditions in the micromagnetic simulations. Validate whether the implicit treatment of dipolar interactions accurately reflects the experimental multilayer structure.

Our reply

We thank the reviewer for these comments. In our simulations, periodic boundary conditions (PBCs) were applied along the in-plane x and y directions to replicate the extended multilayer geometry of our experimental films and to eliminate edge effects. Because the device dimensions in the experiment are much larger than the skyrmion diameter, PBCs allow us to model a representative skyrmion in an effectively infinite film, ensuring that its dynamics remain free of boundary artifacts. To verify the convergence of both dipolar field distributions and skyrmion profiles, we employed eight repetitions of the multilayer unit along each in-plane axis.

Dipolar interactions were treated using MuMax3's FFT-based magnetostatics solver, which efficiently computes long-range dipole-dipole fields under PBCs. This approach accurately captures both intra-layer and inter-layer stray fields. In particular, for thin (1 nm) and closely spaced multilayers, the FFT-based method reproduces the continuous interlayer stray-field coupling that characterizes our experimental system. This methodology has been extensively applied and validated in prior studies of synthetic antiferromagnetic multilayers.

Reviewer's comment

7. Replace the "Guide to eye" lines with theoretical fits or explicitly label them as trendlines to improve clarity and reduce subjectivity.

Our reply

We thank the reviewer for the suggestion. To remove any subjectivity, we have relabeled the line as "Trendline." In addition, we have connected the experimental data points with lines to demonstrate that the trendline accurately follows the measurements. The revised Fig.3 is shown below, and we believe the visibility has been significantly improved.

[Figure 3]

<To>

Reviewer's comment

8. *Supplementary Data: In Supplementary Figure 1, quantify the stability of AFM coupling before and after current pulsing (e.g., through magnetic contrast analysis) to substantiate claims of robustness.*

Our reply

We thank the referee for this suggestion. To substantiate the robustness of the antiferromagnetic (AFM) coupling in the studied stack, we present Supplementary Figure 1, which provides a quantitative analysis of the magnetic contrast before and after current pulsing. The XMCD (X-ray Magnetic Circular Dichroism) line profiles at both the Fe L₃-edge and Co L₃-edge were extracted from the respective regions (highlighted in yellow in the left and right panels of Supplementary Figure 1a) before and after five successive current pulses. We have revised Supplementary Figure 1 below as suggested by the reviewer.

[Supplementary Figure 1]

<To>

Supplementary Figure 1 | Direct observation of stable antiferromagnetic coupling of skyrmion tubes. **a**, the stack ST2 before (left Fig.) and after (right Fig.) five times the current pulses. The current density of the injected pulse was $J = 9.7 \times 10^{11} \text{ Am}^{-2}$. A perpendicular magnetic field of $\mu_0 H_z = 130 \text{ mT}$ was applied at room temperature. **Panel b and c** denote exemplary contrast profiles along the wire-edge direction for each skyrmion highlighted in yellow in Fig. 1a. **b**, Fe L₃-edge contrast and **c**, Co L₃-edge contrast. Red and blue curves denote measurements taken before and after current pulsing, respectively.

Reviewer's comment

9. References: Ensure that all relevant references, such as *National Science Review*, Vol. 6, No. 2, pp. 210-212 (2019), are cited appropriately within the manuscript to acknowledge prior work and provide context.

Our reply

We sincerely thank the reviewer for the suggestion. We have added the recommended references

at the locations we deemed most appropriate as follows. In our manuscript, we believe we have cited well-considered and relevant literature based on the best of our knowledge. However, if there remain any deficiencies or missing citations, we would be grateful for your further guidance to further improve the quality of our manuscript

[page 3, lines 54-55]

<From>

Topological spin textures that emerge in magnetic materials and behave as quasi-particles⁷⁻¹⁸ have been extensively investigated in the realm of electronics¹⁹⁻²¹ for next-generation devices.

<To>

Topological spin textures that emerge in magnetic materials and behave as quasi-particles⁷⁻¹⁸ have been extensively investigated in the realm of electronics¹⁹⁻²² for next-generation devices.

[References]

<Added>

19. Zhou, Y. Magnetic skyrmions: intriguing physics and new spintronic device concepts. *National Science Review* **6**, 210-212 (2019). <https://doi.org/10.1093/nsr/nwy109>

Reviewer's comment

Addressing these points will significantly enhance the clarity, rigor, and impact of the manuscript.

Our reply

We believe that the revisions implemented during this review process have markedly enhanced the clarity, rigor, visibility and overall impact of the manuscript relative to previous versions.

Response to Reviewer #1

We are grateful to the reviewer for the conscientious reading of our manuscript and the helpful suggestions. We are glad to hear the positive appreciation of our work, and we have revised the manuscript according to the comments and incorporated the suggestions. Our responses are listed below.

Reviewer's comment

Authors answered all my questions and questions from the other reviewer satisfactorily. In my opinion, this manuscript can now be published in Nature Communications.

Our reply

We are pleased to learn that our manuscript has been deemed suitable for publication in Nature Communications. We would like to express our sincere gratitude for the reviewers' thorough and constructive evaluation of our draft. We believe that their detailed feedback has greatly enhanced the quality of our manuscript compared with its original version.

Response to Reviewer #2

We are grateful to the reviewer for the conscientious reading of our manuscript and the helpful suggestions. We are glad to hear the positive appreciation of our work, and we have revised the manuscript according to the comments and incorporated the suggestions. Our responses are listed below.

Reviewer's comment

The authors have satisfactorily addressed most of the major comments raised by the reviewers. Therefore, I recommend that it be published in NC as is.

Our reply

We are pleased to learn that our manuscript has been deemed suitable for publication in Nature Communications. We would like to express our sincere gratitude for the reviewers' thorough and constructive evaluation of our draft. We believe that their detailed feedback has greatly enhanced the quality of our manuscript compared with its original version.